# Dual blockade of the lipid kinase PIP4Ks and mitotic pathways leads to cancer-selective lethality

Mayumi Kitagawa[1], Pei-Ju Liao[1], Kyung Hee Lee[1], Jasmine Wong[1], See Cheng Shang [2], Noriaki Minami[3], Oltea Sampetrean[3], Hideyuki Saya [3], Dai Lingyun [4], Nayana Prabhu[4], Go Ka Diam [4], Radoslaw Sobota[5], Andreas Larsson[4], Pär Nordlund[4,5,6], Frank McCormick[7], Sujoy Ghosh[8,9], David M. Epstein[1,10], Brian W. Dymock[2] & Sang Hyun Lee[1]

Achieving robust cancer-specific lethality is the ultimate clinical goal. Here, we identify a compound with dual-inhibitory properties, named a131, that selectively kills cancer cells, while protecting normal cells. Through an unbiased CETSA screen, we identify the PIP4K lipid kinases as the target of a131. Ablation of the PIP4Ks generates a phenocopy of the pharmacological effects of PIP4K inhibition by a131. Notably, PIP4Ks inhibition by a131 causes reversible growth arrest in normal cells by transcriptionally upregulating *PIK3IP1*, a suppressor of the PI3K/Akt/mTOR pathway. Strikingly, Ras activation overrides a131-induced *PIK3IP1* upregulation and activates the PI3K/Akt/mTOR pathway. Consequently, Ras-transformed cells override a131-induced growth arrest and enter mitosis where a131's ability to de-cluster supernumerary centrosomes in cancer cells eliminates Ras-activated cells through mitotic catastrophe. Our discovery of drugs with a dual-inhibitory mechanism provides a unique pharmacological strategy against cancer and evidence of cross-activation between the Ras/Raf/MEK/ERK and PI3K/AKT/mTOR pathways via a Ras⊣PIK3IP1⊣PI3K signaling network.

[1] Program in Cancer & Stem Cell Biology, Duke-NUS Medical School, Singapore 169857, Singapore. [2] Department of Pharmacy, National University of Singapore, Singapore 117543, Singapore. [3] Division of Gene Regulation, Institute for Advanced Medical Research, Keio University School of Medicine, Tokyo 160-0082, Japan. [4] School of Biological Sciences, Nanyang Technological University, Singapore 138673, Singapore. [5] Institute of Molecular and Cell Biology, A*STAR, Singapore 138673, Singapore. [6] Department of Oncology and Pathology, Karolinska Institutet, Stockholm SE-171 77, Sweden. [7] UCSF Helen Diller Family Comprehensive Cancer Center, San Francisco, CA 94158-9001, USA. [8] Department of Cardiovascular and Metabolic Research, Biomedical Biotechnology Research Institute, North Carolina Central University, Durham, NC 27707, USA. [9] Program in Cardiovascular and Metabolic Disorders and Centre for Computational Biology, Duke-NUS Medical School, Singapore 169857, Singapore. [10] Center for Technology & Development (CTeD), Duke-NUS Medical School, Singapore, 169857, Singapore. Correspondence and requests for materials should be addressed to S.H.L. (email: sanghyun.lee@duke-nus.edu.sg)

The Ras/Raf/MEK/ERK and PI3K/Akt/mTOR signaling pathways are essential for cell survival and proliferation in response to external cues. Mutation of proteins within these pathways are among the most common oncogenic targets in human cancers[1,2], and this has spawned a longstanding effort to develop selective inhibitors of these pathways for cancer therapy. Unfortunately, there is ample evidence that cross-talk or cross-amplification of signaling events occurs between these pathways, which both positively and negatively regulate downstream cellular growth events[3]. Moreover, the antitumor activities of single-agent targeted therapies directed to block these signaling pathways has generally been disappointing with an unintended pathway activation leading to drug resistance[4,5]. This has prompted the testing of multiple targeted therapies in combination in order to inhibit multiple oncogenic dependencies[6,7], however, combined treatment with drugs that target the Ras/Raf/MEK/ERK and PI3K/Akt/mTOR signaling pathways has met with marginal clinical success[8]. Thus, there remains the ultimate goal of identifying targets that mediate resistance and cross-talk between these two central pathways.

Here we show a novel compound with dual-inhibitory properties, named a131, that effectively eliminates Ras-activated cancer cells through mitotic catastrophe, while protecting normal cells and allowing them to retain their proliferative capacity. Notably, we have identified the PIP4K lipid kinase family[9,10] as the target of a131 inhibition and delineated a critical role for PIP4K lipid kinases that differently regulate the cell cycle entry between normal and Ras-activated cancer cells. Furthermore, we provide evidence of a mechanism for cross-activation between the Ras/Raf/MEK/ERK and PI3K/AKT/mTOR pathways via Ras-suppressing PIK3IP1, a suppressor of the PI3K/Akt/mTOR pathway, in Ras-pathway activated cancer cells as well as in clinical samples from patients with colorectal and lung adenocarcinomas. Consequently, Ras-activated cancer cells override a131-induced growth arrest and enter mitosis where a131's ability to de-cluster supernumerary centrosomes in cancer cells[11] effectively eliminates Ras-activated cancer cells through mitotic catastrophe. Together, our results provide novel pharmacological strategies against Ras-pathway activated cancers and a mechanism for cross-activation between the Ras/Raf/MEK/ERK and PI3K/AKT/mTOR pathways via a Ras⊣PIK3IP1⊣PI3K signaling network, which promises further insight into the role of this signaling network in regulating cross-talk known to drive response and resistance to clinically relevant targeted therapies.

## Results

**a131 causes selective killing effects in cancer cells**. We undertook a small-molecule screen to investigate the specific signaling networks needed for the proliferation and survival of transformed cells using isogenic human BJ foreskin fibroblasts either immortalized with only hTert (hereafter named as normal BJ) or fully transformed with hTert, small t, shRNAs against p53 and p16 and H-RasV12-ER (estrogen receptor-fused H-Ras bearing the activating G12V mutation) (hereafter named as transformed BJ)[12]. We identified one of the screened compounds (anticancer compound 131; hereafter referred to as a131) (Fig. 1a) that efficiently killed transformed BJ cells, but not normal counterparts (Fig. 1b; Supplementary Fig. 1a). In contrast, treatment with paclitaxel (microtubule stabilizer) and nocodazole (microtubule destabilizer) showed minimal selectivity (Fig. 1b). FACS analysis of the cell cycle revealed that a131 dramatically induced cell death (<2N) only in transformed BJ cells and not in normal counterparts (Fig. 1c) through apoptosis (Fig. 1d; Supplementary Fig. 1b). Moreover, a131 treatment significantly induced aneuploidy (>4N) only in transformed BJ cells (Fig. 1c, panel d'). Instead,

a131 arrested normal BJ cells at the $G_1$/S phase of the cell cycle with few BrdU incorporation (Fig. 1c, panel b'), which was also confirmed with gene set enrichment analysis (GSEA) of genes promoting the cell cycle (Supplementary Fig. 1c). Importantly, this a131-induced growth arrest in normal BJ cells was transient and reversible after a131 removal (Supplementary Fig. 1d). Further, unlike the DNA-damaging Topo II inhibitor etoposide, a131-induced growth arrest occurred in the absence of genotoxic stresses (Supplementary Fig. 1e). This cancer-selective lethality of a131 was further confirmed using a panel of human normal and cancer cell lines ($GI_{50} = 6.5$ vs. $1.7\,\mu M$ (normal vs. cancer)) (Fig. 1e; Supplementary Fig. 2a; Supplementary Data 1). Of note, the difference in $GI_{50}$ values between normal and cancer cells using MTT assay that measures cell proliferation rate (Fig. 1b, e) is likely underestimated, since a131 preferentially induced cell death in transformed and cancer cells, but not normal cells (Supplementary Fig. 2b), whereas it only arrested normal cells at the $G_1$/S phase of the cell cycle in a transient and reversible manner (Supplementary Fig. 1d). Nonetheless, these data suggest that a131 is a potent antiproliferative agent with a clear selectivity toward cancer cells killing.

We utilized a series of engineered human BJ cell lines[12] to delineate the molecular basis for a131-mediated tumor cell-selectivity (Fig. 1f), and we found that inhibition of the p16-pRb, p53, and PP2A tumor suppressor pathways in various combinations did not significantly contribute to a131-induced cell death. In contrast, we found that 4-OHT-induced acute activation of H-RasV12-ER alone was sufficient to sensitize normal BJ cells to a131-induced cell death, and this effect was further enhanced in the context of transformed BJ cells (Fig. 1f). Together, these data indicate that a131 displays a strong selective lethality against Ras-activated or Ras-transformed cells.

Consistent with a131-induced aneuploidy in transformed cells (Fig. 1c, panel d'), time-lapse analysis revealed that a131 treatment immediately induced mitotic arrest in transformed BJ cells (Supplementary Fig. 2c) concomitant with massively misaligned chromosomes (Fig. 1g; Supplementary Fig. 2d), which subsequently missegregated into daughter cells often with catastrophic multipolar division, leading to cell death (Supplementary Movies 1 & 2). In contrast, such catastrophic cell division rarely occurred in a131-treated normal BJ cells (Supplementary Movies 3 & 4) and demonstrated significantly fewer mitotic defects than those in transformed cells (Supplementary Fig. 2c & d). Detailed analysis using high-resolution immunofluorescence microscopy revealed that a131 caused de-clustered centrosomes and multipolar mitotic-spindles in transformed BJ (Fig. 1g, h) and other cancer cells (Supplementary Fig. 2e), since most cancer cells contain supernumerary centrosomes clustered in a bipolar manner before division[11]. In contrast, functional bipolar spindle formed in a majority of a131-treated normal BJ cells that were able to enter mitosis (Fig. 1h; Supplementary Fig. 2f; Supplementary Movie 4) despite a slight decrease in metaphase spindle length (Supplementary Fig. 2g). Together, these data suggest a131 as a potent antimitotic agent that preferentially kills cancer and transformed cells by inducing immediate cancer-selective mitotic catastrophe in vitro, while it arrests normal cells at the $G_1$/S phase of the cell cycle in a reversible manner, which explains a broad-spectrum of its anticancer effects (Supplementary Fig. 2a & b).

The antitumor activities of a131 and of b5, a derivative of a131 designed to improve aqueous solubility, were further determined in mouse xenograft models derived from both HCT-15 human colon adenocarcinoma cells and MDA-MB-231 human breast tumor cells harboring mutant K-RasG13D. As expected, paclitaxel did not show significant antitumor activity against HCT-15 (Fig. 1i). Whereas, both oral and intraperitoneal injections of

a131 and b5 demonstrated marked antitumor efficacies without any body weight loss (Fig. 1i) and cancer cell death as determined by TUNEL staining (Fig. 1j). As observed in in vitro tissue culture, b5 treatment caused massively misaligned chromosomes with multipolar spindles in tumor sections (Fig. 1k). Moreover, in a tumor spheroid culture or orthotopically implanted ex vivo model, a131 treatment significantly suppressed growth of Ras-driven glioma-initiating cells (GICs) (Supplementary Fig. 3a & b). In addition, a131-induced apoptosis only in tumors, but not surrounding normal tissues in ex vivo model (Supplementary Fig. 3c). Taken together, a131 is a unique compound with a potent and broad anticancer efficacy by inducing cancer-selective mitotic catastrophe in vitro, ex vivo, and in vivo.

**a131 eliminates cancer cells via a dual-inhibitory mechanism.** Using various derivatives of a131, we found that the properties of a131 can be separated pharmacologically into two distinct pharmacophores (experimental details and a summary of the results presented in Supplementary Fig. 4a–c and Supplementary Data 2). Through the analysis of a131 structure-and-activity relationship, we classified these compounds into four groups: Group 1 compounds, which possess the dual-inhibitory properties of both causing the arrest of normal BJ cells at the $G_1/S$ phase and also causing mitotic arrest/catastrophe in transformed BJ cells (e.g., a131, b5); Group 2 compounds, which only cause the arrest of normal BJ cells at the $G_1/S$ phase (e.g., a166); Group 3 compounds, which cause mitotic arrest/catastrophe in

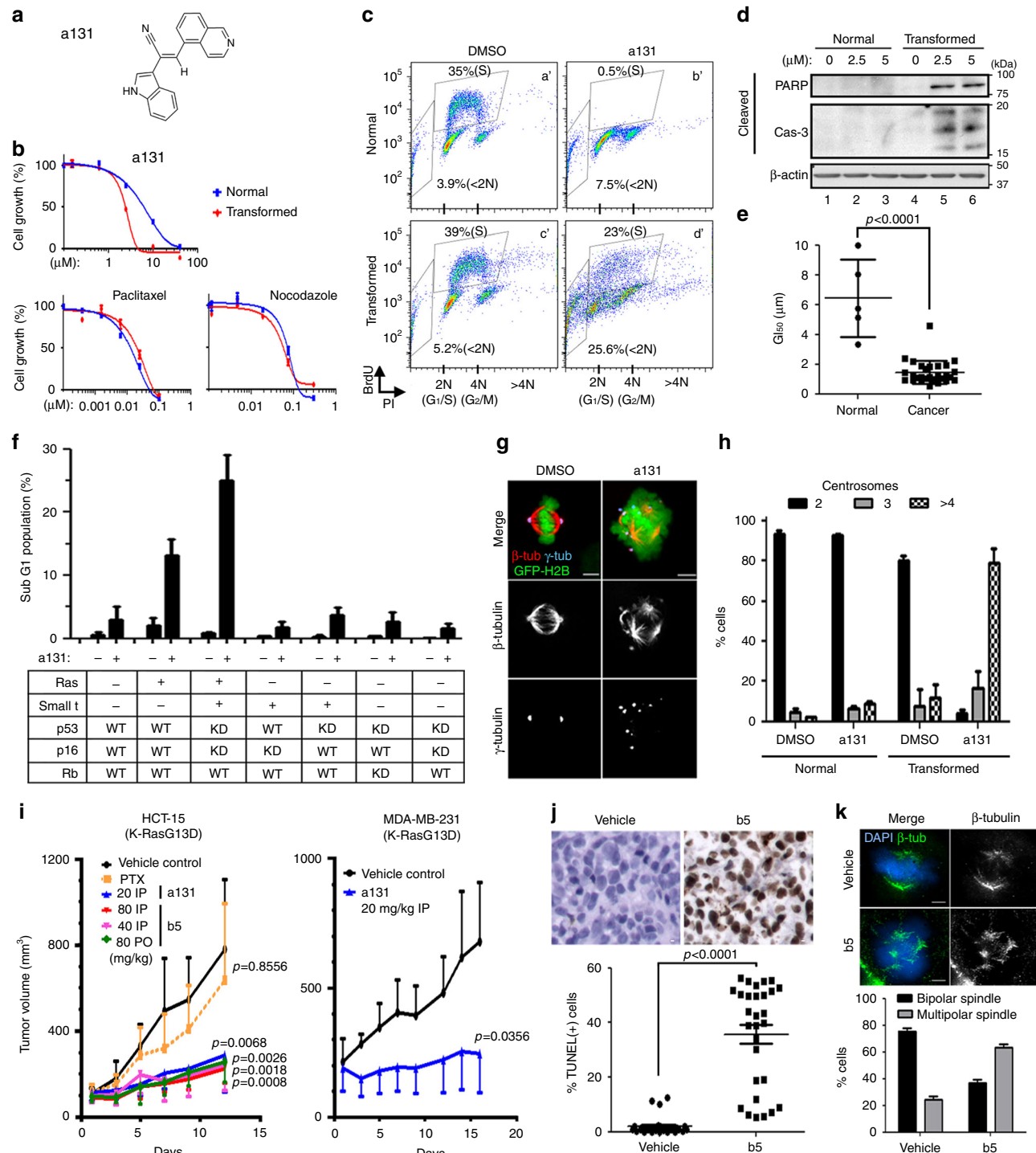

transformed BJ cells, but do not arrest normal BJ cells at the $G_1/S$ phase (e.g., a159); and Group 4 compounds, which are inactive or weakly active (e.g., a132). Importantly, only Group 1 compounds retain the ability to selectively kill transformed BJ cells (Supplementary Fig. 4c). In contrast, Group 2 and Group 4 compounds failed to kill either normal or transformed cell lines, while compounds in Group 3 killed both normal and transformed cell lines with much less selectivity than those in Group 1 (Supplementary Fig. 4c & d; Fig. 1b). Notably, a131-like cancer-selective lethality was recapitulated by combining compounds in Groups 2 and 3 (Supplementary Fig. 4e). Moreover, while paclitaxel and etoposide treatment alone showed a minimal selectivity against transformed BJ cells, pretreatment with a166 in Group 2 markedly augmented such selectivity by protecting normal BJ cells from chemotherapeutic toxicity (Supplementary Fig. 4f & g). Together, these data suggest that the dual-inhibitory property of compounds in Group 1 (e.g., a131) is essential to achieve cancer-selective lethality. Furthermore, compounds in Group 2 (e.g., a166) and Group 3 (e.g., a159) can be classified as chemoprotective and chemotherapeutic agents, respectively.

**Identification of PIP4Ks as target of a131.** To identify cellular targets and signaling pathways of a131 that are responsible for arresting only normal BJ cells at the $G_1/S$ phase of the cell cycle, we explored the mass spectrometry implementation of the cellular thermal shift assay (MS-CETSA) for target identification on the proteome level[13–15]. To increase confidence in target identification, both a131 and a166 were applied for CETSA analysis to find common target proteins. After collecting data covering >8000 proteins in lysates of normal BJ cells, >4000 proteins were used for each compound in the final analyses. Using ranking based on Euclidian distances and thermal shift size, we selected 16 and 11 proteins as potential significant hits for a131 and a166, respectively (Supplementary Fig. 5; Supplementary Data 3; ranking strategies are discussed in Methods). Ferrochelatase in a131 and coproporphyrinogen-III oxidase in a166 were identified as prominent hits. These two proteins of the heme synthesis pathway, however, have previously been identified as promiscuous binders of multiple drugs[15,16], indicating their inhibition is unlikely to give the observed phenotypes of a131 and a166. Instead, the members of PIP4Ks (phosphatidylinositol 5-phosphate 4-kinases)[9,10] stood out as the most prominent common hits, which could constitute candidates for the pharmacological targets; two out of the three

family members (PIP4K2A and 2C) in a131 and all three family members (PIP4K2A, 2B, and 2C) in a166 were identified as CETSA hits (Fig. 2a, b). Indeed, a131 was able to inhibit the kinase activity of purified PIP4K2A in vitro as well as PIP4Ks activity with $IC_{50}$ of 1.9 μM and 0.6 μM, respectively (Fig. 2c, d). Likewise, both a166 and I-OMe-AG-538, previously reported to show PIP4K2A inhibition[17], also inhibited the PIP4K2A activity with $IC_{50}$ of 1.8 and 2.1 μM, respectively (Fig. 2c). a166 also inhibited in vitro PIP4Ks activity, although somewhat less than a131 did (Fig. 2c, d). Of note, a132 in Group 4 compounds, which failed to kill either normal or transformed BJ cell lines (Supplementary Fig. 4c), exerted ~10% inhibition of PI5P4Kα enzyme activity at both ~2 and ~5 μM, and ~40% inhibition at the high concentration of >10μM, although this inhibition was significantly less than that of a131, a166 or I-OMe-AG-538 (Fig. 2c). It is also worth noting that a159 in Group 3 compounds with antimitotic activity exhibited no inhibitory activity against PI5P4Kα in vitro, but it did exhibit some measurable inhibitory activity against the endogenous PI5P4Ks (Fig 2c, d), suggesting a possibility that some of the other proteins in the hit list correspond to proteins that make less but direct physical interactions with the compounds leading to additional off-targets or a result of poly pharmacological effects. Importantly, knockdown of all PIP4K isoforms using three different sets of siRNAs (Supplementary Fig. 6a) induced growth arrest only in normal BJ cells (Fig. 2e), a phenocopy of a131 and a166 treatment (Fig. 1c; Supplementary Fig. 4). The PIP4K genes appear redundant in this assay, since knockdown of individual PIP4K isoforms did not show a significant growth inhibition in normal BJ cells (Supplementary Fig. 6b, c). Moreover, GSEA and KEGG pathway analysis revealed that PIP4Ks knockdown in normal BJ cells downregulated the set of genes promoting cell cycle (Fig. 2f) with a significant number of comparably upregulated or downregulated common cellular pathways as similar to a131 and a166 treatment (Supplementary Fig. 6d, e). Of note, siRNA-mediated knockdown of other CETSA hits including adenosine kinase (ADK) and pyridoxal kinase (PDXK) did not show a significant growth inhibition in normal BJ cells (Supplementary Fig. 7a, b), demonstrating specificity of PIP4Ks as the cellular targets of a131 and a166. Furthermore, similar to a166 treatment (Supplementary Fig. 4e–g) PIP4Ks knockdown also showed significant chemoprotective effects only in normal BJ cells from paclitaxel, etoposide and a159 treatment (Supplementary Fig. 7c–e). Similar to a166 treatment alone (Supplementary Fig. 4c), PIP4Ks knockdown did not induce a significant apoptosis (<2N) in normal BJ cells (Supplementary

**Fig. 1** Selective killing effects of a131 in cancer cells. **a** Structure of a131. **b** Normal and transformed BJ cells were treated with a131, paclitaxel or nocodazole at a range of different concentrations for 72 h in triplicate and cell viability was determined by MTT assay in comparison with indicated antiproliferative agents, hence a reduction in cell viability would imply cell cycle arrest and cell death. Mean values with ± standard deviation (S.D.) are shown ($n = 3$). **c, d** Normal and transformed BJ cells were treated with a131 at 2.5 μM for 48 h. **c** FACS analysis using BrdU and PI double staining of the indicated cells. Percentages of BrdU-positive (S) population indicating cell cycle arrest and subG1 (<2N) population indicating cell death are shown. Representative FACS analysis shown ($n = 3$). **d** Immunoblot analysis of cleaved PARP and caspase-3 (Cas-3) indicating a131-induced apoptosis in transformed BJ cells (lanes 5, 6), but not normal counterparts (lanes 2, 3). Representative immunoblots shown ($n = 3$). **e** MTT assay of human normal and cancer cell lines treated with a131 for 72 h ($n = 3$). Mean concentration values for a131 to achieve 50% growth inhibition ($GI_{50}$) in each cell line are plotted with ± S.D. **f** FACS analysis of a series of engineered BJ-derived fibroblasts treated with a131 at 5 μM for 48 h. Mean values of subG1 (<2N) population ± S.D. are shown ($n > 3$). **g, h** Normal and transformed BJ cells stably expressing GFP-histone H2B were treated with a131 at 2.5 μM for 32 h prior to fixation. **g** Immunofluorescence analysis with representative images ($n = 3$). **h** Quantification of cells ($n > 100$ per condition) with the indicated centrosome numbers. Mean values ± S.D. are shown ($n = 3$). **i** Mice bearing established tumor xenografts were treated orally (PO) or intraperitoneally (IP) twice daily with the indicated doses of a131 or its derivative b5. Tumor volumes were calculated periodically as indicated. Paclitaxel (PTX) was administered intravenously for twice every 4 days. Mean tumor volumes ± S.D. from six mice are shown. Two-way ANOVA was performed to determine statistical significance compared to vehicle control. **j, k** TUNEL staining (**j**) or immunohistochemical analysis (**k**) of HCT-15 tumor sections on Day 12 with representative images (top). **j** The percentage of TUNEL-positive cells were calculated from tumor sections. Note that detection of apoptosis by TUNEL and scanning of the entire sections of each slides, which contained both tumors and adjacent normal tissues, was performed in a completely unbiased manner. Moreover, for unbiased quantification, we used five images from each sections of >6 sections from each group (bottom). **k** Mean values of cell populations ($n > 100$ per condition) with multipolar mitotic-spindles ± S.D. are shown (bottom) with representative images (top). White bars, 5 μm. Where indicated, two-tailed unpaired $t$ tests were performed to determine statistical significance

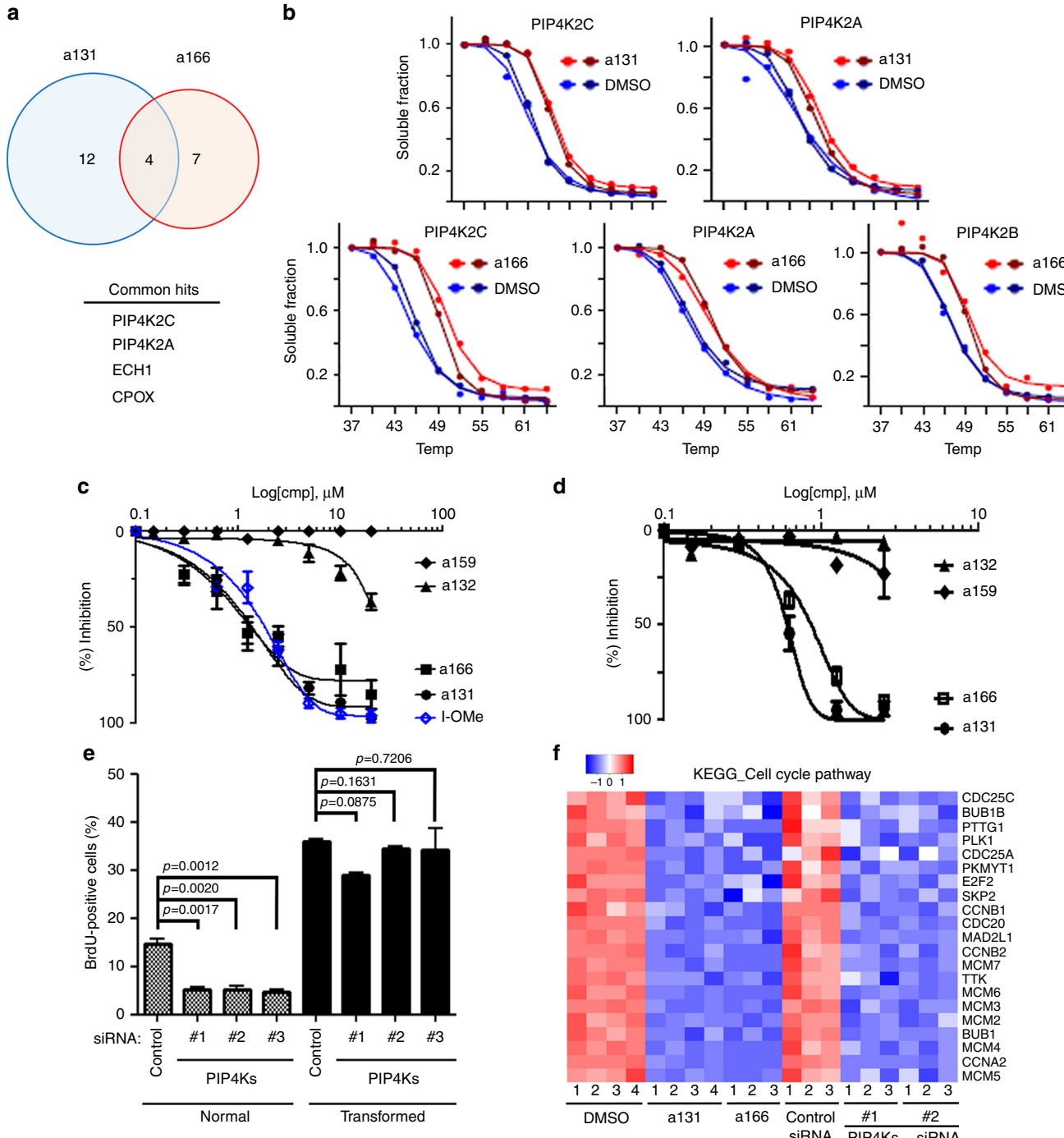

**Fig. 2** Identification of PIP4Ks as targets of a131 responsible for selective growth arrest in normal cells. **a**, **b** Target identification using CETSA. All experiments were performed in two fully independent replicates. **a** Venn diagrams of positive hits from a131 and a166 with the list of commonly targeted hits. Individual hits were ranked by distances (Methods). **b** Melting curves for PIP4K isoforms in duplicated experiments of a131 vs. DMSO (top) or a166 vs. DMSO (bottom) treatment. **c**, **d** PIP4Ks enzyme activity assays using PI5P as substrate (Methods). Inhibition curves ± S.D. ($n = 4$) of the indicated compounds are shown. **c** In vitro PIP4K2A enzyme activity pre-incubated with the indicated compounds. **d** In vitro PIP4Ks activity using HeLa cells in order to verify cell type-independent inhibition by a131 or its indicated derivatives. The compounds were further added to cell lysates at the indicated concentrations during the measurement of PIP4K activity. **e** BrdU incorporation assay as in Fig. 1c. Normal and transformed BJ cells were transfected with control or three different sets of PIP4Ks siRNAs (mixture of PIP4K2A, PIP4K2B, PIP4K2C) for 48 h. The percentage of cells with BrdU-positive populations is shown with mean values ± S.D. ($n = 3$). Two-tailed unpaired $t$ tests were performed to determine statistical significance. Note that the differences in the percentage of BrdU-positive cells between normal and transformed BJ cells were due to a relatively slower growth of normal BJ cells as compared to transformed cells, and not due to a significant growth arrest with the control siRNA. **f** GSEA enrichment plot and heatmap of KEGG cell cycle pathway genes. Normal BJ cells were treated with a131 and a166 at 5 μM for 24 h or transfected with indicated siRNAs for 48 h. The per-sample expression profiles of these genes are depicted in the heatmap using an intensity-based, row-normalized color scale from blue to red, with blue indicating lower expression

Fig. 7e). Likewise, PIP4Ks knockdown in transformed BJ cells only caused a marginal increase in apoptosis (Supplementary Fig 7e), indicating that the dual-inhibitory property of a131 is essential to achieve cancer-selective lethality. Together, these data suggest that PIP4Ks are the cellular targets of a131 and a166. This is to the best of our knowledge the first published study where MS-CETSA has been used to unravel pharmacological targets for hits from a phenotypic screen.

**Ras overrides PI3K pathway suppression by PIP4Ks inhibition.** Recent studies using cancer cells and knockout mouse models of individual PIP4K isoforms[18–22] indicate the involvement of PIP4Ks in controlling the PI3K/Akt/mTOR pathway, well known to promote the $G_1/S$ phase transition. Interestingly, PIP4K loss-of-function mutants in Drosophila possessing only one isoform of PIP4K show inhibition of the PI3K/Akt/mTOR

pathway[23]. Importantly, a131 treatment or PIP4Ks knockdown using three different sets of siRNAs also consistently caused inhibition of the PI3K/Akt/mTOR pathway only in normal BJ cells, but not in transformed counterparts (Fig. 3a, b). Likewise, 4-OHT−induced H-RasV12-ER activation was sufficient to reactivate the PI3K/Akt/mTOR pathway in normal BJ cells even after a131 and a166 treatment or PIP4Ks knockdown (Fig. 3c), which correlates with activated Ras overriding a131-induced, a166-induced, or PIP4Ks knockdown-induced growth arrest in normal BJ cells (Fig. 3d). Together, these data suggest a role of PIP4Ks in promoting the PI3K/Akt/mTOR signaling pathway in a Ras-dependent manner.

**Ras-PI3K pathway cross-talk via Ras⊣PIK3IP1⊣PI3K network.** The molecular components that control the interactions between the Ras/Raf/MEK/ERK and the PI3K/Akt/mTOR pathways is not

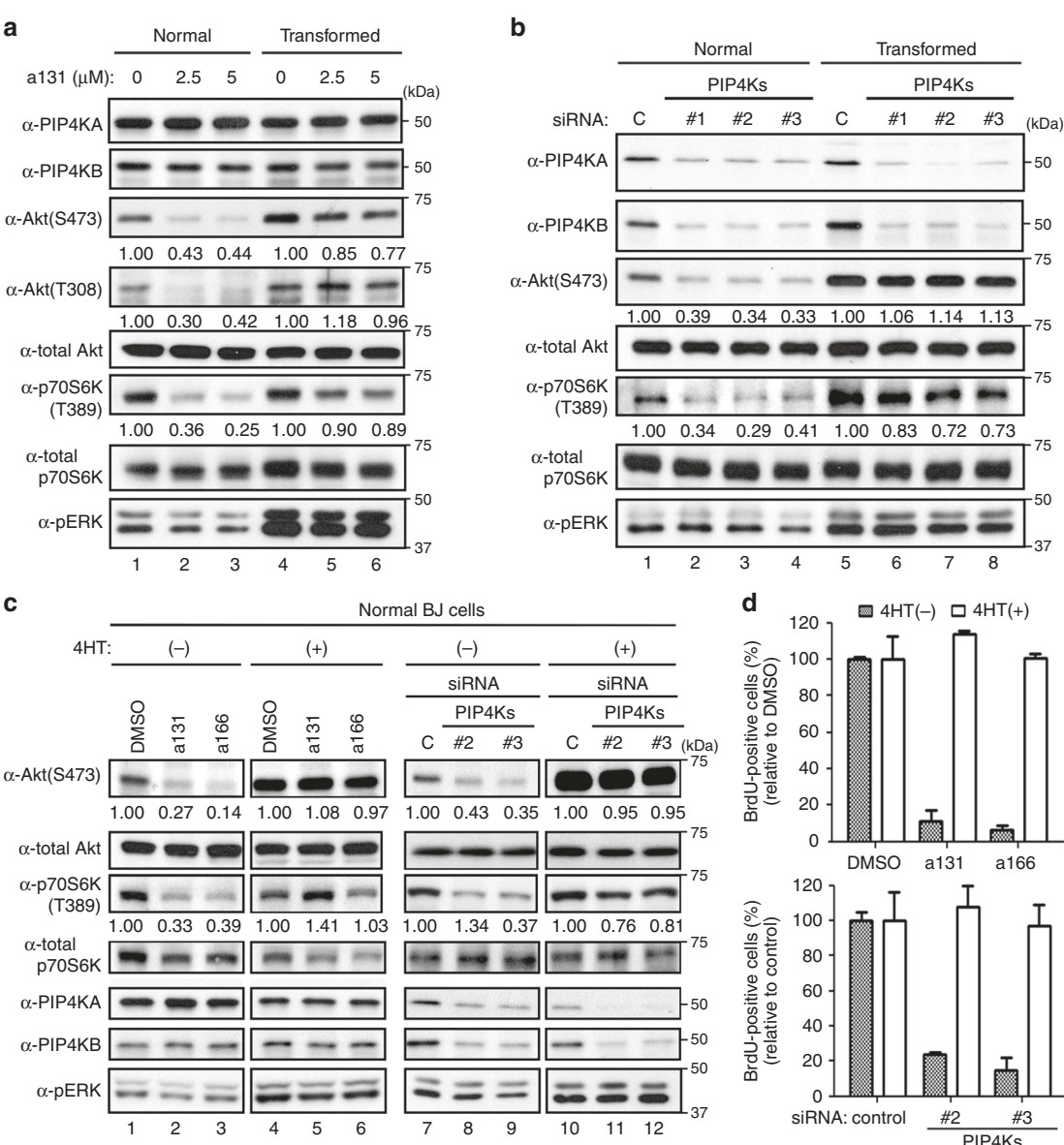

**Fig. 3** a131 and Ras antagonistically control the PI3K/Akt/mTOR pathway. **a–c** Immunoblot analysis of normal and transformed BJ cells treated with a131 or a166 for 24 h **a**, **c** or transfected with indicated siRNAs for 48 h **b**, **c**. Relative ratios of phosphorylated/total levels of Akt and p70S6K are shown in comparison with DMSO. 4HT(+) indicates normal BJ cells treated with 4-OHT for 24 h to activate H-RasV12-ER. Representative immunoblots shown (n = 3). **d** BrdU incorporation assay as in Fig. 1c. 4-OHT [4HT(+)] was added in normal BJ cells for 24 h. Subsequently, cells were treated with a131 or a166 at 5 μM for 48 h (top). Normal BJ cells were transfected with indicated siRNAs for 48 h and treated with 4-OHT for 24 h (bottom). The percentage of cells with BrdU-positive population in comparison with DMSO is shown with mean values and ± S.D. (n = 3)

fully understood[3]. Neither a131 and a166 treatment nor PIP4Ks knockdown inhibited the Ras/Raf/MEK/ERK pathway in normal BJ cells, as determined by ERK phosphorylation (Fig. 3a–c). Thus, to determine how a131 controls the PI3K/Akt/mTOR pathway in a Ras-dependent manner, we interrogated differences in the gene expression levels of known regulators and effectors associated with PI3K in normal and transformed BJ cells upon a131 treatment. Strikingly, among these genes, the PI3K-interacting protein 1 gene (*PIK3IP1*) was significantly upregulated only in a131-treated normal BJ cells (Fig. 4a). Indeed, qRT-PCR and immunoblot analysis confirmed upregulation of PIK3IP1 at both the mRNA and protein levels in either a131-treated and a166-treated or PIP4Ks knockdown normal BJ cells (Fig. 4b, c). Conversely, *PIK3IP1* mRNA expression in transformed BJ cells was not only significantly lower, but also was unresponsive to a131 and a166 treatment (Fig. 4b). Moreover, 4-OHT-induced H-RasV12-ER activation was sufficient to downregulate mRNA and protein levels of PIK3IP1 in a131-treated normal BJ cells (Fig. 4c) and to

dissociate RNA polymerase II (Pol II) from the *PIK3IP1* promoter (Fig. 4d). In contrast, pharmacological inhibition of MEK-attenuated H-RasV12-ER-induced *PIK3IP1* suppression (Fig. 4e), suggesting the molecular basis for positive cross-talk between the Ras/Raf/MEK/ERK and PI3K/Akt/mTOR pathways is mediated by negative transcriptional regulation of *PIK3IP1*.

**High MAPK activity suppresses *PIK3IP1* in human cancers.** PIK3IP1 binds the p110 catalytic subunit of PI3K heterodimers and inhibits PI3K catalytic activity, which leads to inhibition of the PI3K/Akt/mTOR pathway, and *PIK3IP1* dysregulation contributes to carcinogenesis[24–27]. Therefore, we determined whether a131-mediated upregulation of *PIK3IP1* was indeed responsible for the observed inhibition of the PI3K/Akt/mTOR pathway and the $G_1$/S phase transition in normal BJ cells. Indeed, PIK3IP1 knockdown in normal BJ cells significantly restored activation of the PI3K/Akt/mTOR pathway and rescued the population of

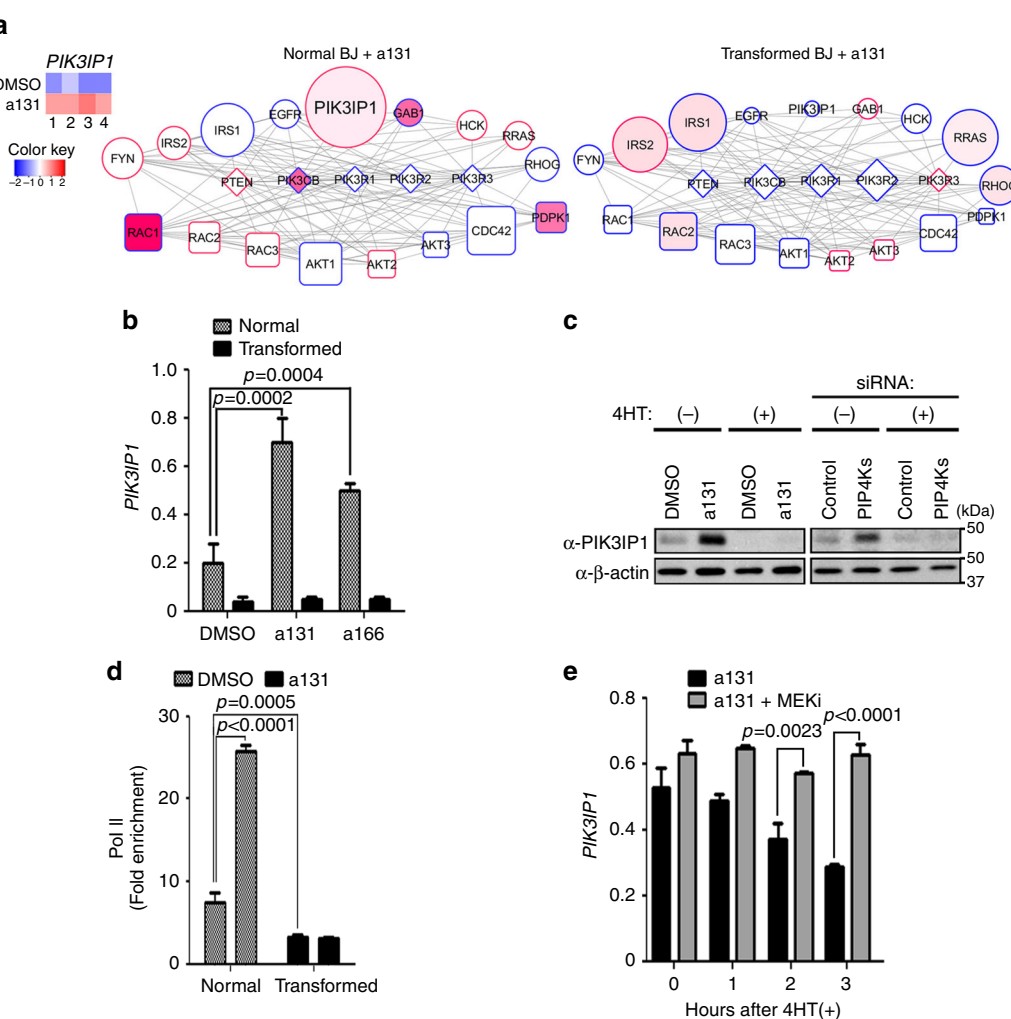

**Fig. 4** Identification of positive cross-talk between Ras/Raf/MEK/ERK and PI3K/Akt/mTOR pathway via Ras⊣PIK3IP1⊣PI3K signaling network. **a** The PI3K network analysis using gene expression of PI3K regulators in normal (middle) and transformed BJ cells (right) treated with a131 at 5 μM for 24 h. PI3K regulators, including *PIK3IP1*, were identified to interact with PI3K with experimental support and high confidence from STRING. The per-sample expression profiles of PIK3IP1 are depicted in the heatmap (left). Color: negative log FDR (false discovery rate), coded from white to red in a scale from 0.15 to 5.37. Size: log ratio; Border: upregulation (red) or downregulation (blue); Shape: upstream (square), parallel (diamond) or downstream (circle). **b, e** qRT-PCR analysis of PIK3IP1 expression in BJ cell lines treated with a131 or a166 at 5 μM for 24 h. Mean values ± S.D. (n = 3). **c** Immunoblot analysis of normal BJ cells treated with 4-OHT [4HT(+)] for 24 h to activate H-RasV12-ER and subsequently with a131 (left) or transfected with indicated siRNAs for 48 h (right). Representative immunoblots shown (n = 3). **d** ChIP analysis of normal BJ cells treated with a131 using antibodies against RNA polymerase II (Pol II) for the *PIK3IP1* gene promoter (n = 3). **e** Normal BJ cells treated with a131 were subsequently treated with MEK inhibitor U0126 (10 μM) for additional 2 h. Then, 4-OHT added to activate H-RasV12-ER for various time points. Where indicated, two-tailed unpaired t tests were performed to determine statistical significance

BrdU-positive proliferative cells, which were suppressed by a131 treatment (Fig. 5a, b). Together, these data reveal positive cross-talk between the Ras and PI3K pathways via the Ras⊣PIK3IP1⊣PI3K signaling network. Moreover, *PIK3IP1* mRNA levels were not only considerably lower in Ras-mutant and Raf-mutant

cancer cells compared with normal cells (Supplementary Fig. 8a), but a131-mediated and a166-mediated induction of *PIK3IP1* was also significantly attenuated in these cancer cells, unlike normal cells (Fig. 5c; Supplementary Fig. 8b). Similarly, analysis of an Oncomine dataset derived from patient samples reveals that

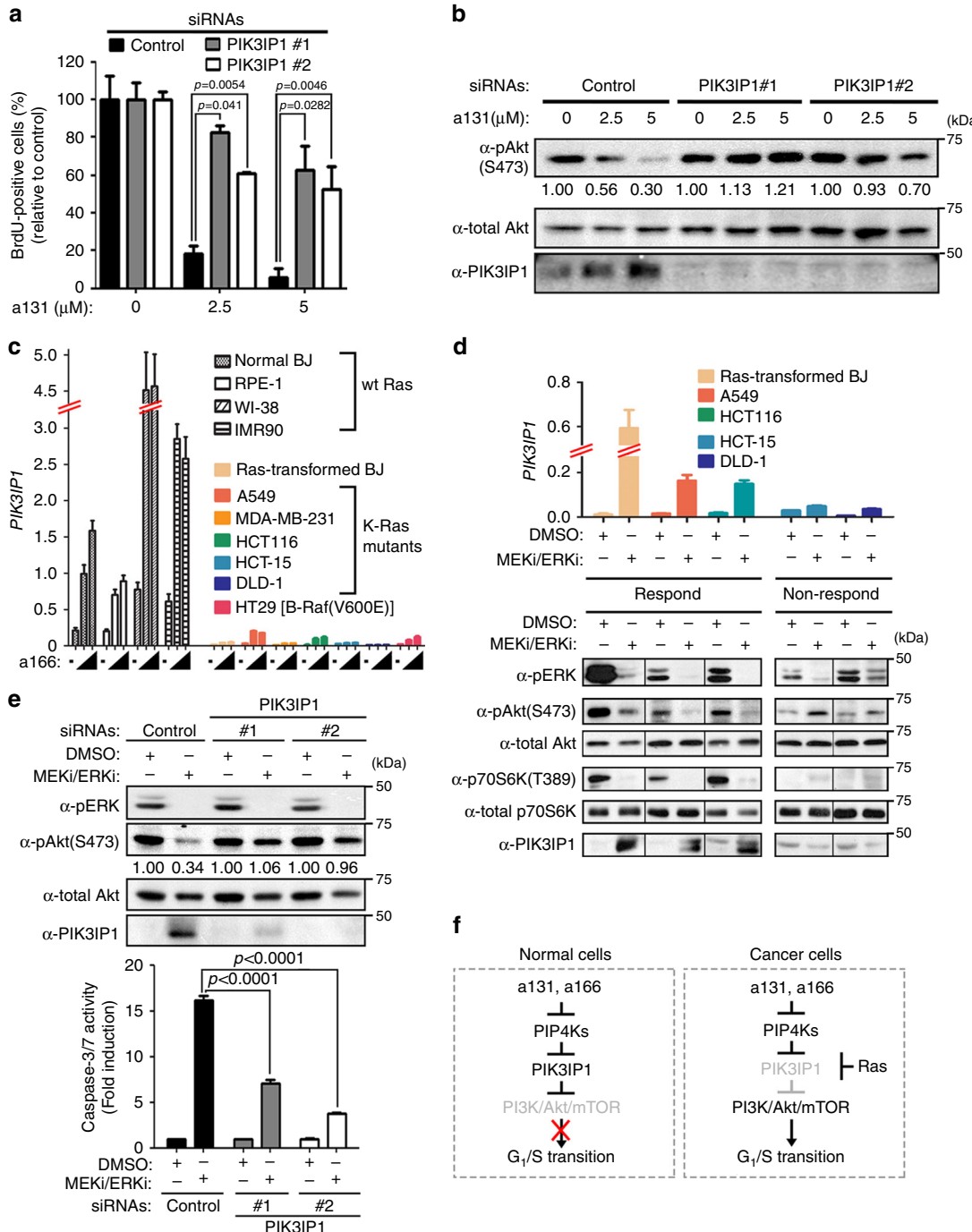

**Fig. 5** The high MAPK activity suppresses *PIK3IP1* in human cancers. **a, b** BrdU incorporation (*n* = 3) (**a**) and immunoblot analysis (**b**) of normal BJ cells transfected with indicated siRNAs for 48 h and treated with a131 for additional 24 h. **c** qRT-PCR analysis of PIK3IP1 expression. Various normal and Ras-mutant cancer cell lines were treated with a166 at 0, 2.5, and 5 μM for 24 h. Mean values ± S.D. (*n* = 3). **d** qRT-PCR analysis of *PIK3IP1* expression (top) and immunoblot analysis (bottom) of Ras-transformed and Ras-mutant cells treated with MEK inhibitor U0126 (10 μM) and ERK inhibitor SCH722984 (1.2 μM) for 24 h. **e** Immunoblot analysis (top) and caspase-3/7 activation (bottom, *n* = 3) of HCT116 cells transfected with indicated siRNAs for 48 h and treated with U0126 (10 μM) and SCH722984 (1.2 μM) for additional 24 h. **f** Schematic model of cross-talk between Ras/Raf/MEK/ERK and PI3K/Akt/mTOR pathway via Ras⊣PIK3IP1⊣PI3K signaling network for cancer cell proliferation. Where indicated, relative ratios of phosphorylated/total levels of Akt and p70S6K are shown compared with DMSO. **b, d, e** Representative immunoblots shown (*n* = 3). Where indicated, two-tailed unpaired *t* tests were performed to determine the statistical significance

*PIK3IP1* expression was significantly lower in human colorectal and lung adenocarcinomas where Ras mutations and activation of Ras signaling pathways are common compared with their corresponding normal tissues or squamous cell lung carcinoma where Ras mutations are uncommon (Supplementary Fig. 8c). Indeed, a negative correlation between *PIK3IP1* expression and Ras mutation status in human colorectal and lung adenocarcinomas were observed (Supplementary Fig. 8d). Conversely, pharmacological inhibition of MEK and ERK significantly increased *PIK3IP1* expression in many Ras-mutant and Raf-mutant cancer cells (Fig. 5d; Supplementary Fig. 8e), while this observed de-repression of *PIK3IP1* was much prominent in most of Raf-mutant cancer cells (Supplementary Fig. 8e), indicating the high MAPK activity is responsible for the suppression of *PIK3IP1*. Furthermore, this de-repression of *PIK3IP1* correlated with concomitant inhibition of the PI3K/Akt/mTOR pathway in HCT116, A549 and transformed BJ cells, but not in those cells unable to de-repress *PIK3IP1* (Fig. 5d). Conversely, PIK3IP1 knockdown significantly restored activation of the PI3K/Akt/mTOR pathway and suppressed cell death induced by inhibition of MEK and ERK (Fig. 5e), further indicating positive cross-talk between the Ras and PI3K pathways via the Ras⊣PIK3IP1⊣PI3K signaling network for cancer cell proliferation and survival (Fig. 5f).

## Discussion

The detailed mechanism of how a131 and activated Ras antagonistically regulate *PIK3IP1* expression requires further investigation. PI(5)P is known to interact with its receptors possessing PHD fingers (e.g., ING2, TAF3) in order to control selective gene expression[22,28]. Interestingly, 20-fold increase in nuclear PI(5)P was observed during the $G_1$/S phase of the cell cycle[29]. Although it remains to be determined, given that a131 treatment or PIP4Ks knockdown also arrested normal cells at the $G_1$/S phase, it is tempting to speculate that a131 treatment or PIP4Ks knockdown may increase PI(5)P levels to activate a nuclear receptor that promotes transcriptional upregulation of *PIK3IP1*. Conversely, activated Ras may signal to inhibit such nuclear receptors or transcription factors to suppress *PIK3IP1* expression, thereby establishing positive cross-talk with the PI3K pathway in Ras-pathway mutated/activated cancers for proliferation. Given that MEK and ERK inhibitors attenuated Ras-mediated suppression of *PIK3IP1*, the Raf/MEK/ERK cascade is likely involved in inhibiting such transcriptional receptor, which is an important topic for further detailed investigation.

The requirement of PIP4Ks for tumorigenesis, especially in the absence of p53, has been demonstrated[18]. Although the mitotic targets of a131 responsible for de-clustering supernumerary centrosomes in cancer cells remain to be determined, together with a131's ability to inhibit PIP4Ks, its potent and broad anticancer efficacy by inducing cancer-selective mitotic catastrophe (Fig. 6) provides novel pharmacological strategies against not only Ras-pathway mutated/activated cancers, but more broadly applicable to a vast majority of human cancers. Furthermore, our discovery of a mechanism for cross-activation between the Ras/Raf/MEK/ERK and PI3K/AKT/mTOR pathways via a Ras⊣PIK3IP1⊣PI3K signaling network promises further insight into the role of this signaling network in regulating cross-talk known to drive response and resistance to clinically relevant targeted therapies.

## Methods

**Cell lines and culture and reagents**. Isogenic BJ human foreskin fibroblast cell lines, including non-transformed (normal) and transformed BJ cells and all gastric cancer cell lines, were kind gifts from Dr. Mathijs Voorhoeve[12] and Dr. Patrick Tan (Duke-NUS)[30], respectively, and tested for mycoplasma infection. The culture

media for the cell lines used in this study are summarized in Supplementary Data 1. All other human cancer cell lines used in this study were purchased from ATCC and cultured in accordance with ATCC's instructions. H-RasV12-ER was activated by exposing the BJ-derived fibroblasts to 4-OHT (100 nM, Sigma-Aldrich). Three different sets of siRNAs were used in this study to target PIP4K isoforms: pool#1, PIP4K2A (5′-CTGCCCGATGGTCTTCCGTAA-3′), PIP4K2B (5′-CACGATCAA TGAGCTGAGCAA-3′), and PIP4K2C (5′-CCGGAGCAGTATGCTAAGCGA); pool#2, PIP4K2A (5′-CGGCTTAATGTTGATGGAGTT-3′), PIP4K2B (5′-CCC TCGATCTATTTCCTTCTT-3′), and PIP4K2C (5′-CCGGAGCAGTATGCTAA GCGA-3′); pool#3, PIP4K2A (5′-CCTCGGACAGACATGAACATT-3′), PIP4K2B (5′-CAAACGCTTCAACGAGTTTAT-3′), and PIP4K2C (5′-CCGAGTCAGTGT GGACAACGA). To knockdown PIK3IP1, siRNAs #1 (5′-AGAGGCTAACCTG-GAAACTAA-3′) and #2 (5′-TACACTGTTATTCATGGTTAA-3′) were used. To knockdown ADK, siRNA (5′-GACACAAGCCCTGCCAAAGAT-3′) was used. To knockdown PDXK, siRNA (5′-CGAAGGCTCGATGTACGTCC-3′) was used. Non-silencing control siRNA was purchased from Dharmacon. For siRNA transfection, Lipofectamine 2000 (Invitrogen) or Dharmafect (Dharmacon) was used according to the manufacturer's instructions.

**Cytotoxicity test**. Cells were plated in 96-well microplates on day 0, and a131 was added to each well on day 1 at a range of different concentrations (from 0.1 to 40 μM) in triplicate. After 3 days of culture, the number of viable cells was determined using MTT cell proliferation assays by adding thiazolyl blue tetrazolium bromide (MTT reagent, Invitrogen) at a concentration of 0.5 mg/ml to each well and incubating for 4 h at 37 °C. The medium was then removed, and the blue dye remaining in each well was dissolved in DMSO by mixing with a microplate mixer. The absorbance of each well was measured at 540 and 660 nm using a microplate reader (Benchmark plus, Bio-Rad). Optical density (OD) values were calculated by subtracting the absorbance at 660 nm from the absorbance at 540 nm. Mean OD values from control cells containing only DMSO-treated wells were set as 100% viable. The concentration of drug that reduced cell viability by 50% ($GI_{50}$) was calculated by non-linear regression fit using GraphPad Prism.

**Crystal violet staining**. Cells were washed twice in 1 × PBS and stained with 0.5% crystal violet dye (in methanol:deionized water = 1:5) for 10 min. Excess crystal violet dye was removed by washing five times (10 min per wash) with deionized water on a shaker, and the culture plates were dried overnight.

**Analysis of cell death and cell growth arrest**. Cell death was assessed via Annexin V and/or PI (propidium iodide) staining according to the manufacturer's instructions (eBioscience). Cell growth arrest was assessed by direct measurement of DNA synthesis through incorporation of the nucleoside analog bromodeoxyuridine (BrdU). Briefly, BrdU (30 μM, Sigma-Aldrich) was added for 2 h before harvesting cells. Cells were subsequently stained with Pacific Blue-conjugated BrdU antibody (Invitrogen) for 1 h followed by PI staining. Stained cells were analyzed by MACSQuant (MACS). Three independent experiments were performed in triplicate. The percentage of Annexin V/PI-positive or BrdU-positive cells was quantified using Flow Jo software (Becton Dickinson). Where indicated, the combined activity of caspase-3/7 was determined using the caspase-Glo 3/7 Assay Kit (Promega) and normalized to the number of viable cells as determined by MTT assay.

**In vivo study**. BALB/c athymic female nude mice (*nu/nu*, 5–7 weeks) (InVivos) were kept under specific pathogen-free conditions. The care and use of mice was approved by the Duke-NUS IACUC in accordance with protocol 2015/SHS/1030. HCT-15 human colon cancer cells ($5 \times 10^6$) or MDA-MB-231 human breast cancer cells ($4 \times 10^6$ with Matrigel) were subcutaneously injected into the flanks of mice. When the mean tumor volume reached 100–300 mm$^3$ (Day 1), the mice were randomly divided into experimental groups of 6 mice by an algorithm that moves animals around to achieve the best case distribution to assure that each treatment group has similar mean tumor burden and standard deviation. No statistical method was used to predetermine sample size. The animals were treated with either intraperitoneal (IP) or oral (PO) injection of a131 (20 mg/kg), b5 (40–80 mg/kg) or vehicle control twice per day for 12 days (HCT115) or 15 days (MDA-MB-231). a131 and b5 were dissolved in DMSO followed by the addition of PEG400 and deionized water (pH 5.0) (final concentrations, 10% DMSO, 50% PEG400). Paclitaxel (Cayman Chemical) was dissolved in ethanol:Tween 80 = 1:3 (v/v) solution followed by the addition of a 5% glucose solution (final ratio, ethanol: Tween 80:5% glucose = 5:15:80) and injected via the tail vain (IV). Tumor dimensions were measured using calipers, and tumor volume (mm$^3$) was calculated using the formula width$^2$ × length/2 in a blinded fashion. On Day 12 (HCT15) or Day 16 (MDA-MB-231), mice were sacrificed. Tumors were collected, fixed overnight in 4% paraformaldehyde (PFA) and stored in 70% ethanol. For immunostaining, antigens were retrieved from formaldehyde-fixed, paraffin-embedded tumor tissue sections for 30 min by boiling in sodium citrate buffer (pH 6.0) using a microwave histoprocessor (Milestone). Endogenous peroxidase activity in tissue sections was depleted by treatment with 3% hydrogen peroxide ($H_2O_2$ in 1 × tris-buffered saline (TBS) for 20 min at room temperature. Tumor tissue sections were incubated overnight with anti-β-tubulin (Abcam; 1:100 in 3% BSA/TBS-Tween 20) at 4 °C followed by incubation with goat anti-rabbit FITC-conjugated secondary

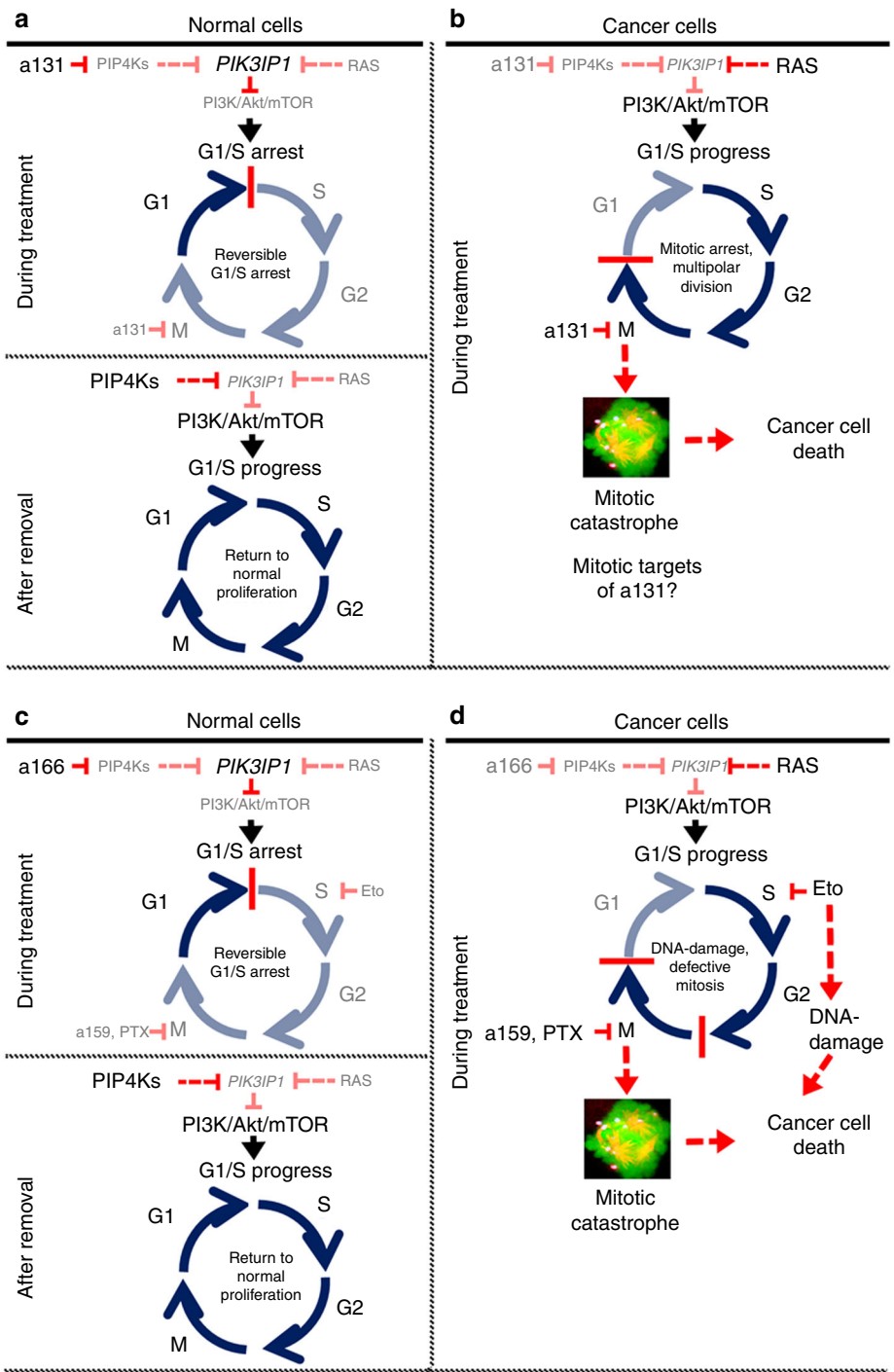

**Fig. 6** Proposed model. Inhibition of PIP4Ks by a131 or a166 arrests normal cells at the G1/S phase of the cell cycle by suppressing the PI3K/Akt/mTOR signaling pathway via transcriptional upregulation of *PIK3IP1* (**a**, top). This cell cycle arrest is reversible after drug removal (**a**, bottom). In contrast, mutation or activation of the Ras/Raf/MEK/ERK pathway in cancer cells promotes positive cross-talk with the PI3K/Akt/mTOR pathway by negative transcriptional regulation of *PIK3IP1*, which allows cancer cells to bypass a131-induced growth arrest at the G1/S phase of the cell cycle, but subsequently leads cancer cells to a131-induced mitotic catastrophe and cell death. However, it is important to note that the mitotic targets of a131 remain unidentified **b**. Pretreatment with a166 protects normal cells from chemotherapeutic toxicity by arresting normal cells at the G1/S phase of the cell cycle **c**. In contrast, mutation or activation of the Ras/Raf/MEK/ERK pathway in cancer cells bypasses the growth arrest, leading to cell death caused by chemotherapeutic drugs (Eto, etoposide; PTX, paclitaxel) and a159 **d**. Of note, this Ras⊣PIK3IP1⊣PI3K signaling network identified in this study may contribute to "oncogene collaboration" of the Ras and PI3K pathways for cancer cell growth and proliferation

antibody (Invitrogen; 1:200 in 3% BSA/TBS) for 1 h at 25 °C. After dehydration treatment, coverslips were mounted using DAPI mounting medium (Vector). Images were acquired in 3D-SIM mode using a super resolution microscope (Nikon)[31], and the number of cells with either 2 or ≥3 mitotic-spindles was quantified (*n* > 50 cells per section, 6–7 sections per treatment). For detection of

apoptosis using the TUNEL method, the ApopTag Plus Peroxidase In Situ Apoptosis Detection Kit (Millipore) was used for formaldehyde-fixed, paraffin-embedded tumor tissue sections treated with b5 (80 mg/kg, IP) or control vehicle for 12 days. Slide scans were acquired using a MetaSystems Metafer built on a Zeiss AxioImager Z.2 upright microscope. The system is equipped with a CoolCube1

camera and a Zeiss Plan-Neofluar 20×/0.5 Ph2 objective lens. Image acquisition was controlled with Metafer4 software, and stitching was performed with the VSlide software and further processed using the open source software FIJI. A custom macro was used to batch process the images in the following sequence: Gaussian filter, color deconvolution of hematoxylin and DAB, thresholding the hematoxylin image, watershed to separate touching nuclei, and then count the number of hematoxylin-stained nuclei. For the DAB image, a fixed threshold was used, watershed applied, and the number of DAB stained nuclei counted. In all cases, no data or animals were excluded and results are expressed as mean and standard deviation of the mean.

**Sphere formation assay.** Murine GICs were established and cultured as described previously[32,33]. Briefly, Ink4a/Arf-null neural stem/progenitor cells were transduced with human H-RasV12 and DsRed and propagated in serum-free Dulbecco modified Eagle medium/F12 (Sigma-Aldrich) supplemented with recombinant epidermal growth factor (PeproTech) and basic fibroblast growth factor (Pepro-Tech) at 20 ng/ml, heparan sulfate (Sigma-Aldrich) at 200 ng/ml and B27 supplement without vitamin A (Invitrogen, Carlsbad, CA). GICs were dissociated and plated in 96-well plates at a density of 100 cells/well. Vehicle (DMSO), temozolomide (Sigma-Aldrich) at 100 μM or a131 at 5 μM were added and sphere formation and size evaluated 7 days after plating. Three plates were prepared for each treatment group, and 30 wells were quantified per plate. Images were acquired on a BZ-X700 inverted fluorescence microscope (Keyence). Quantification was performed by Nikon NIS-element software ($n = 90$).

**Brain slice explants and drug treatment.** Fifty thousand GICs were orthotopically implanted into the forebrain of wild-type mice, and at 7 days post-implantation, brain slice explants were established as previously described[33]. Coronal slices (200 μm) were cultured on Millicell-CM culture plate inserts (Millipore) and treated with vehicle or a131 for 4 days. Images were acquired on an FV10i Olympus confocal microscope (Olympus) and tumor area was quantified by Nikon NIS-element software. Experiments were performed in triplicate. At the end of the experiment (Day 4), slices were fixed overnight in 4% paraformaldehyde, embedded in paraffin and then sectioned at a thickness of 4 μm. Deparaffinized sections were stained with rabbit polyclonal antibody against cleaved caspase-3 (Cell Signaling). Immune complexes were detected using Histofine (Nichirei Biosciences) and ImmPACTDAB (Vector Laboratories). All animal experiments were performed in accordance with the animal care guidelines of Keio University.

**qRT-PCR.** Total RNA was isolated from cultured cells using the RNeasy mini kit (Qiagen). cDNA was synthesized from 1 μg of total RNA using the iScript cDNA synthesis kit (Bio-Rad). qRT-PCR analysis was performed using the iQ SYBR Green Super mix (Bio-Rad) using the following gene-specific primers: human PIP4K2A (5′-AAGAAGAAGCACTTCGTAGCG-3′, 5′-ATGGCTCAGTTCATT-GATCGAG-3′), human PIP4K2B (5′-CCACACGATCAATGAGCTGAG-3′, 5′-TCCTTAAACTTAAAGCGGCTGG-3′), human PIP4K2C (5′-CCGGGAAGC-CAGCGATAAG-3′, 5′-AGCTGCACTAGAAACTCCACA-3′), and human PIK3IP1 (5′-GCTAGGAGGAACTACCACTTTG-3′, 5′-GATGGA-CAAGGAGCACTGTTA-3′). The *TATA-binding protein* gene was used for normalization. All PCR reactions were performed in triplicate.

**Microarray data analysis.** Biotin-labeled cRNA was prepared from 250–500 ng of total RNA using the Illumina TotalPrep RNA Amplification Kit (Ambion Inc.). cRNA yields were quantified with a Agilent Bioanalyzer, and 750 ng of biotin-labeled cRNA was hybridized to the Illumina HT-12 v4.0 Expression Beadchip according to manufacturer's instructions (Illumina, Inc.). Following hybridization, bead chips were washed and stained with Cy3-labeled streptavidin according to the manufacturer's protocol. Dried bead chips were scanned on the Illumina BeadArray Reader confocal scanner (Illumina, Inc.). Gene expression signals obtained after chip scanning were quantile normalized in Partek Genomics Suite v6.6 (Partek Inc.). Genes with a normalized maximum average signal <100 in all groups were considered similar to background and removed from further analysis. Sample outliers were detected via principal component analysis in Partek. Differentially expressed genes were identified via 1-way ANOVA with post hoc contrasts specifying the desired pair-wise comparisons. The magnitude of differential gene expression between 2 groups was expressed as the logarithm of the fold change (base 2), and the statistical significance of differences in gene expression were ascertained by the false discovery rate (FDR). For most analyses, genes with an absolute log fold change >0.58 and FDR <5% were considered significantly differentially expressed. Gene expression profiles across comparator groups were visualized through heatmaps generated via the gplots library in R 3.2.3 using the gplots and RColorbBrewer packages, with genes in rows and treatments in columns. Enrichment graph plots for each gene (represented as bars), which are rank-ordered by their signal-to-noise metric between the control and treated compounds or PIP4Ks knockdown samples. Gene expression values were row-normalized and mapped to a color scale representing an ascending scale of expression signals. In some analyses, the gene expression matrix was subjected to hierarchical clustering by Ward's algorithm[34] prior to the generation of heatmaps. To evaluate the effects

of differential gene expression on biological mechanisms, we performed GSEA[35] using a customized version of the KEGG pathway repository obtained from the Molecular Signatures Database, MSigDB[36]. Biological pathways containing 10–200 genes were considered for analysis, and pathways with FDR < 10% were considered statistically significant.

**Immunoblot analysis.** Total cell lysates were prepared with 1% triton lysis buffer (25 mM Tris HCl (pH 8.0), 150 mM NaCl, 1% triton-X100, 1 mM dithiothreitol (DTT), protease inhibitor mix (Complete Mini, Roche) and phosphatase inhibitor (PhosphoStop, Roche)) and subjected to SDS-PAGE. The following antibodies were used for immunoblotting: anti-β-actin (Sigma-Aldrich), anti-cleaved PARP (Abcam, #ab32064), anti-PIP4K2A (#5527), anti-PIP4K2B (#9694), anti-cleaved caspase-3 (#9664), anti-pHistone H3(Ser10) (#3377), anti-pAkt(S473) (#9271), anti-pAkt(T308) (#13038), anti-total Akt (#9272), anti-p70S6K(T389) (#9234), anti-total p70S6K (#9202), anti-pErk (#4370), anti-γ-Histone H2AX (#9718) (Cell signaling), and anti-PIK3IP1 (Proteintech, #16826-1-AP). The secondary antibodies used were sheep anti-mouse IgG HRP and donkey anti-rabbit IgG HRP (Amersham; 1:2000 dilution). Immunoreactive proteins were visualized using ECL reagent (Amersham). Where indicated, intensities of protein bands were quantified by densitometry (Odyssey V3.0), normalized to their loading controls and then calculated as fold expression change relative to DMSO control. The uncropped scans of the blots were presented in Supplementary Fig. 9.

**Immunofluorescence and time-lapse live-cell imaging.** Immunofluorescence analysis was carried out as described previously[37]. Briefly, cells grown on coverglass-bottom chamber slides (Lab-Tek) were fixed with 4% PFA for 15 min at 25 °C. The fixed cells were permeabilized with 0.5% Triton X-100 and exposed to TBS containing 0.1% Triton X-100 and 2% BSA (AbDil). The following primary antibodies were diluted in PBS containing 1% BSA and 0.1% Triton X-100: anti-γ-tubulin (Sigma-Aldrich, #T6557; 1:1000) and anti-β-tubulin (Abcam, #ab18207; 1:2000). Isotype-specific secondary antibodies (1:500 dilution) coupled to Alexa Fluor 488, 594, or Cy5 (Molecular Probes) were used. Cells were counterstained with DAPI (Thermo Scientific). Images were acquired at RT with 3D-SIM mode using a Super Resolution Microscope (Nikon) equipped with an iXon EM+ 885 EMCCD camera (Andor) mounted on a Nikon Eclipse Ti-E inverted microscope with a CFI Apo TIRF (100×/1.40 oil) objective and processed with the NIS-Elements AR software. For time-lapse live-cell analysis, a Stage Top Incubator with Digital CO₂ mixer (Tokai) was used, and images were acquired at 37 °C[31,37].

**The cellular thermal shift assay.** Target identification was performed by CETSA coupled with quantitative mass spectrometry. In brief, normal BJ cells were lysed by combination of freeze/thaw and mechanical shearing with needle in buffer (50 mM HEPES (pH 7.5), 5 mM β-glycerophosphate, 0.1 mM Sodium Vanadate, 10 mM MgCl₂, 1 mM TCEP, and 1× Protease inhibitor Cocktail). The cell debris was removed by centrifugation at 20,000 g for 20 min at 4 °C. Cell lysates were incubated with 100 μM a131, a166 or DMSO for 3 min at room temperature. Each lysate was divided into 10 aliquots for heat treatment at the respective temperatures for 3 min in a 96-well thermocycler, followed by 3 min at 4 °C. The lysate was centrifuged at 20,000 × g for 20 min at 4 °C and the supernatant was transferred into microtubes for MS-sample preparation.

MS-sample preparation: After lysis, at least 100 μg of the protein (measured with a BCA assay) was subjected to reduction, denaturation and alkylation. Samples were subsequently incubated with sequencing grade LysC (Wako) and trypsin (Promega) for digestion overnight at 37 °C. The digested samples were dried using a centrifugal vacuum evaporator, solubilized in 100 mM TEAB. For each run, 25 μg of the digested protein was labeled for 1 h with 10plexTMT (Pierce). The samples were then quenched with 1 M Tris buffer, pH 7.4. The labeled samples were then pooled together and desalted using a C-18 Sep-Pak (Waters) cartridge and the samples were pre-fractionated into 80 fractions using a High pH reverse phase Zorbax 300 Extend C-18 4.6 mm × 250 mm (Agilent) column and liquid chromatography AKTA Micro (GE) system.

LC-MS analysis: The fractions from the pre-fractionation were pooled into 20 fractions and pooled fractions from each experiment were subjected to mass spectrometry analysis using reverse phase liquid chromatography Dionex 3000 UHPLC system combined with Q Exactive mass spectrometer (Thermo Scientific)[38–40]. The following acquisition parameters were applied: Data Dependent Acquisition with survey scan of 70,000 resolution and AGC target of 3e6; Top12 MS/MS 35,000 resolution (at m/z 200) and AGC target of 1e5; Isolation window 1.6 m/z. Peak lists for subsequent searches were generated using Mascot 2.5.1 (Matrix Science) and Sequest HT (Thermo Scientific) in Proteome Discoverer 2.0 software (Thermo Scientific). The reference protein database used was the concatenated forward/decoy Human-HHV4 Uniprot database.

Hit selection and ranking: Proteins with a high bottom plateau at the highest temperature points were deleted using a cutoff at <0.3 for the average reading of the last 3 temperature points in the control (DSMO-treated) condition[15]. Similarly, proteins, for which a top plateau was not present, were deleted using a cutoff at >0.85 for the average reading of the first three temperature points (in our experience proteins that have already starting melting at ~37 °C are more prone to

give false positives in a shift analysis). Euclidean distance (ED) score of thermal shifts of all the proteins with complete replicates were then calculated as follows:

$$\text{ED score} = \frac{\sum \text{ED}_{\text{inter-treatment}}}{10\sum \text{ED}_{\text{inter-replicate}}}$$

where $\sum \text{ED}_{\text{inter-treatment}}$ is the sum of inter-treatment ED indicating the apparent shift of compound-treated protein melting curve from DMSO-treated one, while $\sum \text{ED}_{\text{inter-replicate}}$ is the sum of inter-replicate ED indicating the noise in similarity of protein melting curves from the same treatment in replicate runs. By definition, larger ED score corresponds to the proteins reproducibly showing a significant shift when compared to control, thus they are potential protein hits targeted by the compound or treatment. ED hit lists were generated for a131 and a166 using a cutoff at median +2.75*MAD (median absolute deviation). $\Delta T_m$ value of thermal shifts were calculated as average deviations between control and treated samples at 0.5 fold change, and proteins with significant positive $\Delta T_m$ value of median +2.75*MAD were selected. The proteins with both significant ED score and significant $\Delta T_m$ value were selected as the final hit lists corresponding to 16 and 11 proteins for a131 and a166, respectively. Melting curves which are flat and have a high plateau at the high temperature edge are less likely to correspond to direct binding[15] and optical inspection suggest that, e.g., Arsenate methyl transferase, albeit giving one of the largest $\Delta T_m$, is less likely to be a significant hit corresponding to direct target binding. The analysis steps including protein melting curve plotting, hits selection and ranking were automated using an in-house-developed script using R programming language[41].

Data normalization and melt curve fitting: The fold-changes of any given protein across the heating temperature range were referenced against the lowest temperature condition (typically 37 °C) (i.e., set a constant value of 1.0 for 37 °C sample point). To perform data normalization for each experimental dataset, a fitting factor vector was first derived. The median fold-changes for all proteins at each of the ten heating temperature points were calculated and fitted into a sigmoidal curve to represent the overall melting trend of the whole proteome. Thus, a 10-element fitting factor vector was calculated by dividing the fitted median value over the original median value at each of the ten temperature points. Furthermore, a single valued scaling factor was introduced to correct for possible differences in baseline signals among different runs after each independent fitting, so that the fitted median value at the lowest temperature for all the different runs would be a common constant value of 1.0. This scaling factor was multiplied to each of the element in the vector of fitting factors to generate a new 10-element vector of normalization factors. Finally, the data normalization was achieved by applying the respective normalization factors in the vector to the protein fold-change values of each temperature points in the respective experimental conditions. The melting curves for individual proteins were subsequently fitted using LL.4() regression modeling function (from "drc" package in R) and plotted in a multiplot format for ease of inspection.

**PIP4K enzyme assay.** HeLa cells were treated with DMSO or compounds for 24 h. Cells were lysed with RIPA buffer (Sigma-Aldrich), and total protein concentrations were measured using a bicinchoninic acid protein assay kit (Thermo Scientific). The compounds were further added to cell lysates at the indicated concentrations during the measurement of PIP4K activity. Next, 10 μg of cell lysate was incubated with 1 μM PI(5)P and 500 nM ATP for 1 h at 37 °C. PIP4K activity was measured by recording luminescent signals (Tecan) using a PIP4KII Activity Assay Kit (Echelon) according to the manufacturer's instructions. For cell-free PIP4K2A activity assays, serial dilutions of compounds were pre-incubated with 1 ng of PIP4K2A (kind gift from Daiichi Sankyo Co., Ltd. and Daiichi Sankyo RD Novare Co., Ltd.) in reaction buffer (50 mM HEPES (pH 7.0), 13 mM MgCl2, 0.005% CHAPS, 0.01% BSA, 2.5 mM DTT) for 1 h at 25 °C. DOPS (80 μM, Avanti polar lipids), PI(5)P (20 μM, Echelon) and ATP (10 μM, Sigma-Aldrich) were added and further incubated for 90 min at room temperature. PIP4K2A activity was measured by recording luminescent signals (Tecan) using an ADP-Glo Kinase Assay (Promega) according to the manufacturer's protocol.

**Chromatin immunoprecipitation.** Chromatin immunoprecipitation (ChIP) assays were carried out using the Magna ChIP A/G Kit (Millipore) according to the manufacturer's instructions. Enrichment of Pol II binding to PIK3IP1 was evaluated by qPCR using 1/10 of the immunoprecipitated chromatin as template and iQ SYBR Green Super mix (Bio-Rad). Primer sequences are available upon request.

**Data availability.** Microarray data that support the findings of this study have been deposited in the National Center for Biotechnology Information Gene Expression Omnibus (GEO) and are accessible through the GEO Series accession number GSE104301. All other relevant data are available from the corresponding author on reasonable request.

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

## Acknowledgements

We thank D. M. Virshup (Duke-NUS) for critical comments, and R. M. Bunte (Duke-NUS) and G. Wright (IMB) for TUNEL analysis. This research was supported by the National Research Fellow (S.H.L.; NRF-RF2010-02) program, NMRC CBRG (S.H.L.; NMRC/CBRG/0091/2015), NMRC platform funding, MOH (P.N., R.S.; IAFCAT2/004/2015) Young Investigator Grant (R.S.) and the Duke-NUS Medical School (S.H.L.; NUS/RECA(PILOT/2013/0006)).

## Author contributions

M.K. and S.H.L. conceived and led the functional studies with the technical assistants from P.-J.L., K.H.L., and J.W. S.C.S. and B.W.D. conceived and led the generation of small molecules. N.M. and O.S. performed the tumor spheroid and ex vivo model experiments under the supervision of H.S. D.L., N. P., G.K.D., R.S., and A. L. performed MS-CETSA under the supervision of P. N. S.G. performed GSEA and KEGG pathway analysis. M.K., F.M., D.M.E., and S.H.L. wrote the manuscript and all authors commented on the manuscript.

## Additional information

**Competing interests:** S.H.L., M.K., S.C.S., and B.W.D. are inventors on a filed patent application covering the discoveries of small molecules presented in this manuscript. The remaining authors declare no competing financial interests.

