## [Peer Review File · Nature Communications]

Reviewers' comments:

Reviewer #1 (Remarks to the Author):

In this study the authors identify and characterise a small molecular weight inhibitor a131 which shows a propensity to kill cancer cells more efficiently than normal cells. The molecule appears to do this based on its ability to induce cell cycle arrest and to induce a mitotic catastrophe, however the cell cycle arrest appears to be lost in cancer cells as a consequence of upregulation of the map kinase pathway. These two phenotypes can apparently be split into different pharmacophore properties, exemplified by a133, which induces both phenotypes, a166, which only appears to induce the cell cycle arrest in wild type cells, and a159, which induces a mitotic catastrophe. By carrying out mass spectrometry coupled with temperature stability the authors identify PIP4K as a potential binder. Further studies show that the activity of PIP4K are indeed inhibited by A166 and that knockdown of PIP4K phenocopies a133 and a166 in terms of cell cycle arrest in normal cells. The authors would like to suggest that the combination of cell cycle arrest in control cells, induced through PIP4K inhibition, compared to cancer cells and the mitotic catastrophe combine to induce the cancer specific selectivity.

In essence the paper is a very interesting study and deserves to be published here if the following criticisms can be addressed.

Major concerns

1. The title of the paper is "Dual Blockade of the Lipid 1 Kinase PI5P4Ks and Mitotic Pathways Leads to Cancer-selective Lethality" however this is not what the study actually show. The author show that they can split the two phenotypes in terms of molecules (a166 induces a cell cycle arrest , while a159 induces the mitotic catastrophe), and that adding the two molecules together re-induces the cancer cell selective phenotype. The obvious but extremely relevant experiment would be to add back a159 to PIP4K knockdown cells and show that this reengages a cancer selective phenotype. Without this data the title cannot be used.

2. The authors do not decipher which PIP4K is required for the phenotype, although they show that they can knockdown individual isoforms. Both mouse and cell data show that these isoforms can act both redundantly and non-redundantly on different phenotypes. Therefore the authors should address which isoform is required.

3. The authors should show whether knock down of PIP4K induces apoptosis.

4. The authors suggest that the inhibitor a133 can inhibit PIP4K in vivo. However this is assessed using in vitro experiment, where the cells are incubated first with the inhibitor and then lysed and PIP4K activity measured by an ATP loss assay. This suggests one of two things,

a. The inhibitor binds irreversibly as there must be a huge dilution of its concentration upon cell lysis.

b. The mechanism of inhibition in vivo is not down solely to binding to the PIP4K but possibly through mechanisms that lead to stable decreased activity. This might include posttranslational mechanism.

The authors should at least convincingly determine whether the inhibition is irreversible.

5. The authors suggest that a159 which induces mitotic catastrophe does not show the cancer cell selectivity that a133 shows. However looking at the dose curve it is clear that the lowest concentrations shown for a159 already induces maximal killing in cancer cells. The authors should provide a dose curve that goes much lower to determine the cancer versus normal control selectivity.

Minor concerns.

1. "This cancer-selective lethality of a131 was further confirmed using a panel of human normal and cancer cell lines [GI50 = 6.5 vs. 1.7 μ M (normal vs. cancer)]" this difference does not appear to be so great. Could the authors please include a discussion about how this might compare to the use of other inhibitors?

2. The very last figure in which the authors add the Ym inhibitor should e removed. It does not add to the paper. There are no data on PtdIns5P measurements in this study and without these

the inhibitor data does nothing to firm the case for PIPK and PtdIns5P. Furthermore, the change from 0.44 to 0.76 as assessed by western blotting requires error bars to determine significance.

Reviewer #2 (Remarks to the Author):

The manuscript by Kitagawa et al describes a new molecule a131 which through inhibition of PI5P4Ks and mitotic pathways selectively kills cancer cells.

The targets of the drug are identified using mass spectrometry in combination with the CETSA assay. The validation work is extensive and well executed, and the suggested mechanism is well elucidated and novel. I am in general in favor of publication after some issues are addressed.

General comment: The manuscript is extremely dense and as it stands now would be not easily accessible to a broader readership. I would suggest to expand sections, and describe in a clear way, also possibly adding a clear scheme, the suggested mechanism of action.

It needs to be made more clear that half the targets responsible for the mechanism are identified. As far as I can deduce the targets responsible for de-clustering supernumerary centrosomes are so far not known? If yes, it should be made clearer in the text and also in the suggested mechanism scheme.

Referencing should be improved. The original CETSA paper by Molina et al Science 2013 is not even cited, also when mentioning the combination of CETSA with MS the paper that did it for the first time, Savitski et Science 2014 should be cited.

The methods part should cite more MS relevant literature. When mentioning TMT a couple of references should be given, as well as when mentioning the Q Exactive.

From the supplement: "Melting

curves which are flat and have a high plateau at the high temperature edge are less likely to correspond to direct binding 10 and optical inspection suggest that e.g. Arsenate methyl transferase, albeit giving one of the largest ΔT_m , is less likely to be a significant hit corresponding to direct target binding." Reference 10 has nothing to do with this sentence, did you mean reference 18 (also repeated once as reference 38)?

Was a normalization procedure of any kind used? If yes please provide a short description.

Reviewer #3 (Remarks to the Author):

The manuscript entitled "Dual Blockade of the Lipid Kinase PI5P4Ks and Mitotic Pathways Leads to Cancer-selective Lethality" by Kitagawa et al represents a very comprehensive study regarding a novel inhibitory compound, a phenotypic screen to elucidate a target and attempts to provide a mechanism of action. In summary, the authors present a large amount of very detailed data and thoroughly backed-up experimental lines to support their conclusions. The authors provide strong evidence that this paper fulfills the criteria for publication in the journal; experimental evidence in the most part is backed up with alternative lines of study or orthogonal assays for support, the authors present a novel inhibitor with dual inhibitory properties that has a selectivity against transformed cells, an observation that is of potential importance to the field of oncology as well of interest in cell cycle progression and signal transduction.

The main points of the paper focus on the use of a specific transformed cell line (with an immortalized control without p53/p16 suppression and constitutively active Ras) which were used to perform a small molecule screen. Hits from this screen could be classed as (1) able to growth arrest both lines and kill the transformed line (a131/b5), (2) growth arrest the control cell line and kill neither (a166), (3) growth arrest both lines and kill both lines (a159) and (4) growth arrest neither line and kill neither (weakly active a132). The compound group characterized by a131

displayed a selective lethality specifically against Ras-transformed cells. A phenotypic screen using MS-CETSA was used to identify a cellular target, PI5P4K. PIK3IP1 was subsequently suggested to mediate the response of a131 in Ras-activated transformed cells being able to override the growth arrest induced by inhibition of the mTOR pathway.

The strongest part of this study in my opinion is the comprehensive analysis and multiple lines of experimental investigation for the majority of the claims in the paper. For example, antitumor activities were further evidenced using mouse xenograft models, a panel of human cancer cell lines, tumor spheroid culture and orthotopically implanted Ras-driven GICs. Again, the activity of a131 against PI5P4K α was determined biochemically as well as from cell lysates, genetically phenocopied and evidenced by gene regulation studies. The weakest part of the study lies in the connection of PI5P4K to PIK3IP1 and the suggested mechanisms of action, which the authors did suggest required further investigation.

There was a relatively small number of questions and some minor points that should be addressed by the authors before publication, listed in no particular order of importance.

1. Figure 1b shows a simple crystal violet stain indicating that a131 is able to kill transformed cells. Extended Fig 1a shows a similar result with paclitaxel and nocodazole controls, would this not be a better main figure? This graph seems to show that at 40 μ M a131 kills normal cells, and whilst this effect is 100% growth arrest in transformed cells using 4 μ M, at this concentration 40% of normal cells were non-viable – is this not significant? In the text the authors state that this compound does not kill non-transformed cells.
2. In Fig 1i the symbols used for vehicle control and PTX data points cannot be distinguished, making it difficult to see the fit lines and error bars.
3. In Fig 1k shows immunohistochemical analysis using the b5 analogue, whereas the text refers to this as a131 (page 5 line 7).
4. In Fig 1j the data quantifying the TUNEL-positive staining for b5 treatment seem to have a very large spread across the full range of values, either clustered very high or low. Is this relevant?
5. From the CETSA screen data the authors validate the exclusion of the best hits, ferrochelatase and CPOX. The PI5P4Ks were chosen as targets based on this – were the other hits considered (eg adenosine kinase, pyridoxal kinase)? Can the authors suggest a reason why the other hits have been discounted?
6. Can the authors explain the data that isn't shown (page 7 lines 27/31)? For line 27 the data referred to i.e. mRNA and protein knockdown of PIK3IP1 do seem to be shown – is there more data that isn't?
7. On page 9 (line 5) the authors suggest that inhibition of PI5P4Ks would increase PI5P levels in the nucleus. Whilst a166 indicated PI5P4K β in the CETSA screen, this was not a hit for a131. Seeing as PI5P4K β is the predominant nuclear isoform can the authors be sure of this suggestion? If the authors are supporting this hypothesis by the use of the PIKfyve inhibitor, perhaps they should measure the predicted nuclear lipid levels?
8. In Fig 2c and d the curve fitting for the datasets seems to be poor for a132, a131 and a166 (c) and a131 (d). In Fig 2d the last two data points for a131 seem to be larger than 100% inhibition? The curve for a166 seems to be missing from Fig 2d. In Fig 2c the curve for a132 is minimal inhibition for the weakly active compound (as expected), whereas in Fig 2c against purified pI5P4K α it seems to give quite good inhibition (upto 50%). Conversely a159 (kills normal and transformed cells) has no activity against PI5P4K α but does have some activity against the endogenous PI5P4Ks. Is there an explanation for this?
9. In Fig 2e there seems to be significant growth arrest with the control siRNA. The PI5P4K knockdown is significant but is there a reason why the basal level is so low?
10. In extended Fig 5a CETSA melt curves the y axis is labelled non-denatured protein fraction but all the curves are 0-1, is this correct? Should the control treatment be the same for all of the samples i.e. the blue curve similar for each plot? There seem to be significant differences for example with the Q9H5X1 sample. The authors also suggest that the plot for arsenite methyltransferase is not good due to a high plateau. This also seems to be the same with the B4DN88 sample (with a higher DMSO control curve). Also there seem to be minimal differences

between the control and experimental curves for the last three plots Q13011, P36551 and Q8N684-3 – are these significant?

11. In Fig 2b the plots for a131/a166 PI5P4K α and PI5P4K γ seem to be duplicates from the data presented in extended Fig 5. Should the curve for a166 PI5P4K β also be included?

12. The video file "Mitotic progression of normal BJ cells treated with a131" does not seem to work (downloaded from zip or independently).

In conclusion this is a very well-constructed paper that has an important message, and by addressing the above points would be suitable for publication.

Responses to Referees' Comments:

Reviewer #1 (Remarks to the Author):

In this study the authors identify and characterise a small molecular weight inhibitor a131 which shows a propensity to kill cancer cells more efficiently than normal cells. The molecule appears to do this based on its ability to induce cell cycle arrest and to induce a mitotic catastrophe, however the cell cycle arrest appears to be lost in cancer cells as a consequence of upregulation of the map kinase pathway. These two phenotypes can apparently be split into different pharmacophore properties, exemplified by a133, which induces both phenotypes, a166, which only appears to induce the cell cycle arrest in wild type cells, and a159, which induces a mitotic catastrophe. By carrying out mass spectrometry coupled with temperature stability the authors identify PIP4K as a potential binder. Further studies show that the activity of PIP4K are indeed inhibited by A166 and that knockdown of PIP4K phenocopies a133 and a166 in terms of cell cycle arrest in normal cells. The authors would like to suggest that the combination of cell cycle arrest in control cells, induced through PIP4K inhibition, compared to cancer cells and the mitotic catastrophe combine to induce the cancer specific selectivity.

In essence the paper is a very interesting study and deserves to be published here if the following criticisms can be addressed.

Response to the general comment: We appreciate for the reviewer's very positive comment on our manuscript. We agree with the reviewer comment that in essence the paper is a very interesting study and deserves to be published here. Similarly, the other two reviewers also commented, "The validation work is extensive and well executed, and the suggested mechanism is well elucidated and novel. I am in general in favor of publication" and "In conclusion this is a very well-constructed paper that has an important message". Based on the positive and constructive suggestions made by the reviewer, we have performed a series of additional analysis in order to address all remaining issues. Furthermore, we have carefully revised the manuscript according to the reviewer's comments. In summary, we believe that virtually all comments made by the reviewer have been experimentally addressed and/or carefully discussed in the revised manuscript. Thus, we hope you to find that the revised manuscript is suitable for publication in Nature Communications.

Major concerns:

Comment #1: The title of the paper is “Dual Blockade of the Lipid 1 Kinase PI5P4Ks and Mitotic Pathways Leads to Cancer-selective Lethality”, however this is not what the study actually show. The author show that they can split the two phenotypes in terms of molecules (a166 induces a cell cycle arrest, while a159 induces the mitotic catastrophe), and that adding the two molecules together re-induces the cancer cell selective phenotype. The obvious but extremely relevant experiment would be to add back a159 to PIP4K knockdown cells and show that this reengages a cancer selective phenotype. Without this data the title cannot be used.

Response #1: We appreciate the reviewer’s comment. As the reviewer suggested, we performed the experiments of adding back a159 to PI5P4Ks knockdown normal and transformed BJ cells. As shown in the revised Supplementary Fig 7e, a131-like cancer-selective lethality was recapitulated by PI5P4Ks knockdown and combining a159. These results are consistent with the results shown in the original Supplementary Fig. 4d and revised Supplementary Fig. 4e where a131-like cancer-selective lethality was achieved by combining compounds in Groups 2 and 3. Moreover, these results are also consistent with the results shown in the original Supplementary Fig. 6d & 6e where PI5P4Ks knockdown showed a significant chemoprotective effects only in normal cells from paclitaxel and etoposide, which is presented in the revised Supplementary Fig 7c and 7d. Together, we conclude that dual inhibition of PI5P4Ks and mitotic pathways can lead to cancer-selective lethality.

Comment #2: The authors do not decipher which PIP4K is required for the phenotype, although they show that they can knockdown individual isoforms. Both mouse and cell data show that these isoforms can act both redundantly and non-redundantly on different phenotypes. Therefore, the authors should address which isoform is required.

Response #2: As the reviewer asked to determine whether the reversible growth inhibitory phenotype in normal BJ cells by depleting PI5P4K isoforms occurs in a redundant or non-redundant manner. As the reviewer suggested, we depleted individual PI5P4K isoforms using isoform specific siRNAs. The efficient knockdown of individual PI5P4K isoforms was confirmed using qRT-PCR analysis (Revised Supplementary Fig. 6a & b). Notably, as shown in the revised Supplementary Fig. 6b & 6c, knockdown of individual PI5P4K isoforms did not show a significant growth inhibition in normal BJ cells. In contrast, knockdown of all PI5P4K isoforms (Original Supplementary Fig. 6c; Revised Supplementary

Fig. 6b & 6c) induced growth arrest in normal BJ cells, which is consistent with the data shown in the Original Fig. 2e. Furthermore, we noticed that an individual knockdown of PI5K4K appears to affect the mRNA levels of other PI5P4K isoforms (Revised Supplementary Fig. 6b), indicating a possible redundant or compensatory mechanism. These results are also consistent with our observation that knockdown of all PI5P4K isoforms is required for a robust growth arrest in normal BJ cells. Together, we conclude that PI5P4Ks function redundantly to control cell cycle progression in normal BJ cells. These results are clearly indicated on the page 7 in the revised manuscript.

Comment #3: The authors should show whether knock down of PIP4K induces apoptosis.

Response #3: As the reviewer suggested, we determined whether knockdown of PI5P4Ks induces a significant apoptosis in normal and transformed BJ cells. As shown in the revised Supplementary Fig. 7e, knockdown of all PI5P4K isoforms did not induce a significant apoptosis (<2N) in normal BJ cells, whereas it caused a robust growth arrest in normal BJ cells (Revised Supplementary Fig. 6c). Similarly, knockdown of PI5P4Ks in transformed BJ cells did not cause a significant increase in apoptosis (Revised Supplementary Fig 7e). These results are also consistent with the finding that a166 did not induce a significant apoptosis (Original Supplementary Fig. 4c, d). Together, we conclude that PI5P4Ks largely control the G1/S phase of the cell cycle in normal BJ cells, but not apoptosis. These results are clearly indicated on the page 8 in the revised manuscript.

Comment #4: The authors suggest that the inhibitor a133 can inhibit PIP4K in vivo. However this is assessed using in vitro experiment, where the cells are incubated first with the inhibitor and then lysed and PIP4K activity measured by an ATP loss assay. This suggests one of two things,

- a. The inhibitor binds irreversibly as there must be a huge dilution of its concentration upon cell lysis.
- b. The mechanism of inhibition in vivo is not down solely to binding to the PIP4K but possibly through mechanisms that lead to stable decreased activity. This might include posttranslational mechanism.

The authors should at least convincingly determine whether the inhibition is irreversible.

Response #4: We appreciate the reviewer's comment and apologize for not clearly stating about the addition of the compounds in cell lysates during the in vitro PI5P4K enzyme activity. In the original and revised Fig. 2d, the in vitro PI5P4K enzyme activity not only pre-

incubated with the indicated compounds, but we added the compounds in cell lysates at the indicated concentrations during the measurement of PI5P4K enzyme activity. Furthermore, as shown in the original and revised Supplementary Fig. 1d, the a131-induced growth arrest in normal BJ cells was transient and reversible after a131 removal by wash-out approach. Together, these results strongly suggest that a131 inhibits PI5P4K α enzyme activity in a reversible manner. Nonetheless, we apologise for not stating this clearly in the original manuscript. As the reviewer suggested, we clearly indicated this on the page 17 in the revised manuscript and on the page 10 (PI5P4K enzyme assay) in the revised Supplementary information. We also noted the reviewer intriguing comment, “the mechanism of inhibition in vivo is not down solely to binding to the PIP4K but possibly through mechanisms that lead to stable decreased activity. This might include posttranslational mechanism”. Although we agree with the reviewer’s comment on a potential posttranslational mechanism of controlling PI5P4K activity in vivo, we feel that it is beyond the scope of our current version of manuscript, but hope very much to be part of a future paper.

Comment #5: The authors suggest that a159 which induces mitotic catastrophe does not show the cancer cell selectivity that a133 shows. However looking at the dose curve it is clear that the lowest concentrations shown for a159 already induces maximal killing in cancer cells. The authors should provide a dose curve that goes much lower to determine the cancer versus normal control selectivity.

Response #5: As the reviewer suggested, we added a dose curve of titrating a159 at the various concentration in normal and transformed BJ cells. As shown in the revised Supplementary Fig. 4d, similar to the treatment with other anti-mitotic agents including paclitaxel (microtubule stabilizer) and nocodazole (microtubule destabilizer), a159 showed a minimal selectivity against transformed BJ cells (Revised Fig. 1b), which is consistent with our data demonstrating anti-mitotic activity of a159. In contrast, we demonstrated that only Group 1 compounds (e.g. a131) retain the ability to markedly and selectively kill transformed BJ cells, while compounds in Group 3 (e.g. a159) killed both normal and transformed cell lines with much less selectivity than those in Group 1 (Original and revised Supplementary Fig. 4c). These results are stated on the page 5 (lines 18-21) in the revised manuscript. Please also note that we did not state that a159 does not have any cancer cell selectivity in the original manuscript. Instead, we demonstrated that a159 showed much less cancer cell selectivity than a131 did (Original Supplementary Fig. 4b & 4c). A certain degree of cancer cell selectivity lethality by a159 is expected since a131 and a159 caused more frequent

mitotic catastrophe in transformed BJ and other cancer cells with supernumerary centrosomes than normal cells via de-clustering centrosomes and multipolar mitotic-spindles. However, a key finding described in our original manuscript is that the additional ability of a131 to arrest normal cells at the G1/S and thereby protect normal cells from entering mitosis is required to achieve the maximum cancer selective lethality. We clearly indicated this point on the page 6 in the revised manuscript.

Minor concerns:

Comment #1. “This cancer-selective lethality of a131 was further confirmed using a panel of human normal and cancer cell lines [GI₅₀ = 6.5 vs. 1.7 μM (normal vs. cancer)]” this difference does not appear to be so great. Could the authors please include a discussion about how this might compare to the use of other inhibitors?

Response #1: We appreciate the reviewer’s comment that the difference in the cancer-selective lethality does not appear to be so great. The GI₅₀ values in normal vs. cancer cells from MTT assay were measured after continuously treating normal and cancer cell line at a range of different concentrations of a131 (from 0.1 μM to 40 μM) for 72 h. However, it is important to note that a131 dramatically induced cell death only in transformed and cancer cells, but not in normal counterparts (Original and revised Fig. 1c, 1d; Original Supplementary Fig. 1b; Revised Fig. 1b). Instead of inducing cell death, a131 arrested normal cells at the G1/S phase of the cell cycle, which was transient and reversible after a131 removal (Original and revised Supplementary Fig. 1d). Therefore, these results strongly suggest that a131 is a potent antiproliferative agent with a clear selectivity toward cancer cells killing. Together, we believe that the difference of GI₅₀ values in normal vs. cancer cells does not fully reveal the robust cancer-specific cell lethality induced by a131. We clearly indicated these implications on the page 4 (lines 23-27) in the revised manuscript.

Comment #2: The very last figure in which the authors add the Ym inhibitor should be removed. It does not add to the paper. There are no data on PtdIns5P measurements in this study and without these the inhibitor data does nothing to firm the case for PIPK and PtdIns5P. Furthermore, the change from 0.44 to 0.76 as assessed by western blotting requires error bars to determine significance.

Response #2: We appreciate the reviewer’s comment. The reviewer #3 also pointed out this issue on his/her comment #7. We agree with both reviewers’ comments that the original Fig. 4j using PIKfyve inhibitor YM-201636 does not add much significance to the

scope of our current manuscript. Therefore, as the reviewer suggested, we removed the original Fig. 4j and revised the manuscript accordingly.

Reviewer #2 (Remarks to the Author):

The manuscript by Kitagawa et al describes a new molecule a131 which through inhibition of PI5P4Ks and mitotic pathways selectively kills cancer cells.

The targets of the drug are identified using mass spectrometry in combination with the CETSA assay. The validation work is extensive and well executed, and the suggested mechanism is well elucidated and novel. I am in general in favor of publication after some issues are addressed.

Response to the reviewer's general remark: We appreciate the reviewer's favourable and constructive comment. Similarly, the other two reviewers also indicated very positive comments on our manuscript and their supports for publication by addressing a few remaining points. Based on the reviewer comments, we have carefully revised the manuscript to be more accessible to a broader readership.

General comment:

Comment #1: The manuscript is extremely dense and as it stands now would be not easily accessible to a broader readership. I would suggest to expand sections, and describe in a clear way, also possibly adding a clear scheme, the suggested mechanism of action.

Response #1: We agree with the reviewer's comment that the original manuscript is extremely dense and as it stands now would be not easily accessible to a broader readership. As the reviewer suggested, we have divided and expanded sections with 6 main figures. We also carefully described the results in a clearer way throughout the revised manuscript using Word track changes. Furthermore, in order to make our manuscript more accessible to a broader readership, as the reviewer suggested, we have also added a clear scheme of the suggested mechanism of action by relocating the original Supplementary Fig. 8 to the revised Fig. 6.

Comment #2: It needs to be made more clear that half the targets responsible for the mechanism are identified. As far as I can deduce the targets responsible for de-clustering supernumerary centrosomes are so far not known? If yes, it should be made clearer in the text and also in the suggested mechanism scheme.

Response #2: We appreciate the reviewer's comment. The reviewer correctly pointed out that the targets responsible for de-clustering supernumerary centrosomes are so far not identified. As the reviewer suggested, we clearly indicated the unidentified mitotic targets in the suggested mechanism (Revised Fig. 6.) and described on the page 21 in the revised

manuscript by adding the statement that “it is important to note that the mitotic targets of a131 remain unidentified”. Furthermore, we clearly indicate this in the discussion by adding the statement that “Although the mitotic targets of a131 responsible for de-clustering supernumerary centrosomes in cancer cells remain to be determined, together with a131’s ability to inhibit PI5P4Ks, its potent and broad anticancer efficacy by inducing cancer-selective mitotic catastrophe provides novel pharmacological strategies against not only Ras-pathway mutated/activated cancers, but more broadly applicable to a vast majority of human cancers” on the page 11 in the revised manuscript.

Comment #3: Referencing should be improved. The original CETSA paper by Molina et al Science 2013 is not even cited, also when mentioning the combination of CETSA with MS the paper that did it for the first time, Savitski et Science 2014 should be cited.

Response #3: We apologize for missing and incorrectly citing references. As the reviewer suggested, we added the original CETSA paper by Molina et al Science 2013 and cited Savitski et al Science 2014 when we mentioned the combination of CETSA with MS in the revised manuscript.

Comment #4: The methods part should cite more MS relevant literature. When mentioning TMT a couple of references should be given, as well as when mentioning the Q Exactive.

Response #4: We appreciate the reviewer’s comment. As the reviewer suggested, we cited more following MS relevant literatures in the supplementary methods section, especially for TMT and Q Exactive mass spectrometer:

Tandem Mass Tags: A Novel Quantification Strategy for Comparative Analysis of Complex Protein Mixtures by MS/MS

Andrew Thompson, Jürgen Schäfer, Karsten Kuhn, Stefan Kienle, Josef Schwarz, Günter Schmidt, Thomas Neumann, and, and Christian Hamon

Analytical Chemistry 2003 75 (8), 1895-1904

Ion Coalescence of Neutron Encoded TMT 10-Plex Reporter Ions

Thilo Werner, Gavain Sweetman, Maria Fälth Savitski, Toby Mathieson, Marcus Bantscheff, and Mikhail M Savitski

Analytical Chemistry 2014 86 (7), 3594-3601

Optimized Fast and Sensitive Acquisition Methods for Shotgun Proteomics on a Quadrupole Orbitrap Mass Spectrometer

Christian D. Kelstrup, Clifford Young, Richard Lavalley, Michael L. Nielsen, and Jesper V. Olsen

Journal of Proteome Research 2012 11 (6), 3487-3497

Comment #5: From the supplement: “Melting curves which are flat and have a high plateau at the high temperature edge are less likely to correspond to direct binding 10 and optical inspection suggest that e.g. Arsenate methyl transferase, albeit giving one of the largest T_m , is less likely to be a significant hit corresponding to direct target binding.” Reference 10 has nothing to do with this sentence, did you mean reference 18 (also repeated once as reference 38)?

Response #5: We apologize for incorrectly citing the reference. We corrected this mistake in the revised supplemental information. In addition, we stated more clearly about “Hit selection and ranking” by modifying the first two sentences as follows in the revised supplemental information:

Hit selection and ranking: *Proteins with a high bottom plateau at the highest temperature points were deleted using a cutoff at <0.3 for the average reading of the last 3 temperature points in the control (DSMO-treated) condition. Similarly, proteins, for which a top plateau was not present, were deleted using a cutoff at >0.85 for the average reading of the first three temperature points (in our experience proteins that have already starting melting at $\sim 37^\circ\text{C}$ are more prone to give false positives in a shift analysis).*

Comment #6: Was a normalization procedure of any kind used? If yes please provide a short description.

Response #6: As the reviewer suggested, we provided a short description of normalization procedure as follows in the revised supplemental information:

Data normalization and melt curve fitting: *The fold-changes of any given protein across the heating temperature range were referenced against the lowest temperature condition (typically 37°C) (i.e., set a constant value of 1.0 for 37°C sample point). To perform data normalization for each experimental dataset, a fitting factor vector was first derived. The median fold-changes for all proteins at each of the ten heating temperature points were calculated and fitted into a sigmoidal curve to represent the overall melting trend of the whole proteome. Thus, a 10-element fitting factor vector was calculated by dividing the fitted*

median value over the original median value at each of the ten temperature points. Furthermore, a single valued scaling factor was introduced to correct for possible differences in baseline signals among different runs after each independent fitting, so that the fitted median value at the lowest temperature for all the different runs would be a common constant value of 1.0. This scaling factor was multiplied to each of the element in the vector of fitting factors to generate a new 10-element vector of normalization factors. Finally, the data normalization was achieved by applying the respective normalization factors in the vector to the protein fold-change values of each temperature points in the respective experimental conditions. The melting curves for individual proteins were subsequently fitted using $LL.4()$ regression modeling function (from “drc” package in R) and plotted in a multiplot format for ease of inspection.

Reviewer #3 (Remarks to the Author):

The manuscript entitled “Dual Blockade of the Lipid Kinase PI5P4Ks and Mitotic Pathways Leads to Cancer-selective Lethality” by Kitagawa et al represents a very comprehensive study regarding a novel inhibitory compound, a phenotypic screen to elucidate a target and attempts to provide a mechanism of action. In summary, the authors present a large amount of very detailed data and thoroughly backed-up experimental lines to support their conclusions. The authors provide strong evidence that this paper fulfils the criteria for publication in the journal; experimental evidence in the most part is backed up with alternative lines of study or orthogonal assays for support, the authors present a novel inhibitor with dual inhibitory properties that has a selectivity against transformed cells, an observation that is of potential importance to the field of oncology as well of interest in cell cycle progression and signal transduction.

The main points of the paper focus on the use of a specific transformed cell line (with an immortalized control without p53/p16 suppression and constitutively active Ras), which were used to perform a small molecule screen. Hits from this screen could be classed as (1) able to growth arrest both lines and kill the transformed line (a131/b5), (2) growth arrest the control cell line and kill neither (a166), (3) growth arrest both lines and kill both lines (a159) and (4) growth arrest neither line and kill neither (weakly active a132). The compound group characterized by a131 displayed a selective lethality specifically against Ras-transformed cells. A phenotypic screen using MS-CETSA was used to identify a cellular target, PI5P4K. PIK3IP1 was subsequently suggested to mediate the response of a131 in Ras-activated transformed cells being able to override the growth arrest induced by inhibition of the mTOR pathway.

The strongest part of this study in my opinion is the comprehensive analysis and multiple lines of experimental investigation for the majority of the claims in the paper. For example, antitumor activities were further evidenced using mouse xenograft models, a panel of human cancer cell lines, tumor spheroid culture and orthotopically implanted Ras-driven GICs. Again, the activity of a131 against PI5P4K α was determined biochemically as well as from cell lysates, genetically phenocopied and evidenced by gene regulation studies. The weakest part of the study lies in the connection of PI5P4K to PIK3IP1 and the suggested mechanisms of action, which the authors did suggest required further investigation. There was a relatively small number of questions and some minor points that should be addressed by the authors before publication, listed in no particular order of importance.

Response to the general comment: We appreciate for the reviewer's very positive comment on our manuscript. We agree with the reviewer comment that in essence the paper is a very interesting study. Similarly, the other two reviewers also indicated very positive comments on our manuscript and their supports for publication by addressing a few remaining points. Based on the positive and constructive suggestions made by the reviewer, we have performed a series of additional analysis in order to address all remaining issues.

Comment #1: Figure 1b shows a simple crystal violet stain indicating that a131 is able to kill transformed cells. Supplementary Fig 1a shows a similar result with paclitaxel and nocodazole controls, would this not be a better main figure? This graph seems to show that at 40 μ M a131 kills normal cells, and whilst this effect is 100% growth arrest in transformed cells using 4 μ M, at this concentration 40% of normal cells were non-viable – is this not significant? In the text the authors state that this compound does not kill non-transformed cells.

Response #1: We agree with the reviewer's suggestion that the original Supplementary Fig 1a showing a similar result with paclitaxel and nocodazole controls would be a better main figure than the original Fig. 1b. As the reviewer suggested, we replaced the original Fig. 1b with the original Supplementary Fig. 1a in the revised manuscript. The reviewer also correctly pointed out that treatment of a131 at 4 μ M concentration caused 100% reduction of cell proliferation rate of transformed cells, while we observed that it did so ~20-30% of normal cells using MTT assay. However, we feel it is important to note that the MTT assay measures cell proliferation rate rather than specifically cell death. Importantly, using PI-staining and immunoblot analysis to measure cell death per se, we clearly demonstrated that a131 dramatically induced cell death only in transformed cells, but not in normal counterparts (Original Fig. 1c, 1d). Instead, a131 arrested normal BJ cells at the G1/S phase of the cell cycle, which was transient and reversible after a131 removal (Original Fig. 1c; Original Supplementary Fig. 1d). Therefore, a131 is a potent antiproliferative agent with a clear selectivity toward cancer cells killing, while it allows normal cells to retain their proliferative capacity. Likewise, we believe that ~ ~20-30% reduction of normal cells by a131 treatment at 4 μ M concentration for 72 h as compared to control DMSO treatment (Original Supplementary Fig. 1a) was largely due to a transient and reversible growth arrest in normal cells (original Supplementary Fig. 1d) instead of cell death. We clearly indicated this implication by stating that "Of note, the difference in GI₅₀ values between normal and cancer cells using MTT assay that measures cell proliferation

rate (Fig. 1b & 1e) is likely underestimated, since a131 preferentially induced cell death in transformed and cancer cells, whereas it only arrested normal cells at the G₁/S phase of the cell cycle in a transient and reversible manner” on the page 4 in the revised manuscript.

Comment #2: In Fig 1i the symbols used for vehicle control and PTX data points cannot be distinguished, making it difficult to see the fit lines and error bars.

Response #2: We appreciate the reviewer’s comment and apologize for not distinguishing clearly the symbols used for vehicle control and PTX data points. We corrected this issue in the revised Fig. 1i.

Comment #3: In Fig 1k shows immunohistochemical analysis using the b5 analogue, whereas the text refers to this as a131 (page 5 line 7).

Response #3: We apologize for this mistake in the text. We corrected this issue on the page 5 in the revised manuscript.

Comment #4: In Fig 1j the data quantifying the TUNEL-positive staining for b5 treatment seem to have a very large spread across the full range of values, either clustered very high or low. Is this relevant?

Response #4: We appreciate the reviewer’s comment. We believe that the large spread in a clustered way was due to the fact that detection of apoptosis by TUNEL and scanning of the entire sections of each slides, which contained both tumors and adjacent normal tissues, was performed in a completely unbiased manner, and it thereby may cause some variations of the number of TUNEL positive cells depending on the tumor tissue sections. Moreover, for unbiased quantification, we used 5 images from each sections of >6 sections from each group, which might also reflect heterogeneity of each tumor sections. Therefore, we do not feel any relevance of the large spread of data points since cancer cell death was evidently and statistically significantly determined in an unbiased manner using TUNEL staining and scanning as presented in the original Fig. 1j.

Comment #5: From the CETSA screen data the authors validate the exclusion of the best hits, ferrochelatase and CPOX. The PI5P4Ks were chosen as targets based on this – were the other hits considered (eg adenosine kinase, pyridoxal kinase)? Can the authors suggest a reason why the other hits have been discounted?

Response #5: We appreciate the reviewer's comment. We apologize for not including the data demonstrating that the other two hits including adenosine kinase and pyridoxal kinase were not responsible for cell cycle arrest in normal BJ cells. As included in the revised Supplementary Fig. 7a & 7b, unlike knockdown of PI5P4Ks isoforms, knockdown of those hits did not show significant cell cycle arrest in normal BJ cells, and thereby we excluded these targets for further studies. We indicated this on the pages 7 and 8 in the revised manuscript. Nonetheless, we cannot completely exclude a possibility that some of the other proteins in the hit list correspond to proteins that make direct physical interactions with the compounds, i.e. are off-targets or a result of poly-pharmacology. Alternatively, these hits could be due to effects on downstream pathways. However, when several of the PI5P4K come up as hits and when the shift sizes of these are in general, larger than for other potential hits, we have focused our follow up validation on the PI5P4Ks. The hypothesis is therefore that these are the most prominent targets (although we cannot exclude less prominent targets), which is supported by the observation that knockouts of these proteins capture essential features of the compounds.

Comment #6: Can the authors explain the data that isn't shown (page 7 lines 27/31)? For line 27 the data referred to i.e. mRNA and protein knockdown of PIK3IP1 do seem to be shown – is there more data that isn't?

Response #6: We appreciate the reviewer's comment. We also apologize for the misstatement made on the page 7 in the original manuscript. As the reviewer indicated, we presented all results of immunoblot analysis of PIK3IP1 levels and the qRT-PCR analysis confirmed either up-regulation or down-regulation of PIK3IP1 (Original Fig. 4c) in the original manuscript. Thus, we deleted/corrected the misstatement on the page 9 in the revised manuscript.

Comment # 7: On page 9 (line 5) the authors suggest that inhibition of PI5P4Ks would increase PI5P levels in the nucleus. Whilst a166 indicated PI5P4K β in the CETSA screen, this was not a hit for a131. Seeing as PI5P4K β is the predominant nuclear isoform can the authors be sure of this suggestion? If the authors are supporting this hypothesis by the use of the PIKfyve inhibitor, perhaps they should measure the predicted nuclear lipid levels?

Response #7: We appreciate the reviewer's comment. The reviewer #1 also pointed out this on his/her minor comment #2 and suggested us to remove the original Fig. 4j because the results from using PIKfyve inhibitor YM-201636 do not add any significance to the scope of our current manuscript. As the reviewer also pointed out, we removed the

statements on the page 9 (lines 5-7 & 11-13) in the original manuscript, “Nonetheless, inhibition of PI5P4Ks by a131 and a166 is expected to increase PI(5)P levels in the nucleus, where PI(5)P is known to interact with its receptors possessing PHD fingers (e.g., ING2, TAF3) in order to control selective gene expression” and “which is further supported by our observation that blocking PI(5)P synthesis with the PIKfyve inhibitor YM-201636 attenuated a131-mediated PIK3IP1 expression and restored Akt activation (Fig. 4j)”, and we revised the manuscript accordingly.

Comment #8: In Fig 2c and d the curve fitting for the datasets seems to be poor for a132, a131 and a166 (c) and a131 (d). In Fig 2d the last two data points for a131 seem to be larger than 100% inhibition? The curve for a166 seems to be missing from Fig 2d. In Fig 2c the curve for a132 is minimal inhibition for the weakly active compound (as expected), whereas in Fig 2c against purified PI5P4K α it seems to give quite good inhibition (up to 50%). Conversely a159 (kills normal and transformed cells) has no activity against PI5P4K α but does have some activity against the endogenous PI5P4Ks. Is there an explanation for this?

Response #8: We apologize for the poor curve fitting for the datasets and missing data points for a131 in the original Fig. 2c. We corrected these issues in the revised Fig. 2c. As the reviewer suggested, we also added the curve for a166 in the revised Fig. 2d, which also inhibited combined cellular PI5P4Ks, although somewhat less than a131 did. We clearly described these findings on the page 7 in the revised manuscript. We also agree with the reviewer’s comment that in the original and revised Fig. 2c, the curve for a132 is minimal inhibition for the weakly active compound, whereas in Fig 2c against purified PI5P4K α , a132 showed some measurable inhibition but only at the high concentration of >10 μ M. Conversely, as shown in the original and revised Fig. 2d, a159 (group 3 with anti-mitotic activity) has no activity against PI5P4K α in vitro but does have some activity against the endogenous PI5P4Ks. Although we cannot completely exclude a possibility that some of the other proteins in the hit list correspond to proteins that make less but direct physical interactions with the compounds, i.e. are off-targets or a result of poly-pharmacology. Moreover, obtaining perfect positive correlation between in vitro and in vivo PI5P4Ks assays might be complicated. For instance, for measurement of the in vivo total PI5P4Ks activity, since a159 also possesses anti-mitotic activity and alters the cell cycle progression, this might affect the intrinsic activity of PI5P4Ks even after cell lysates were prepared. Alternatively, these variations could be due to effects on downstream pathways in cell lysates. Nonetheless, we feel that our validation on the PI5P4Ks and phenocopies of a133 and a166

in terms of cell cycle arrest in normal cells as well as gene regulation studies strongly support the observation that knockouts of these proteins capture essential features of the compounds.

Comment #9: In Fig 2e there seems to be significant growth arrest with the control siRNA. The PI5P4K knockdown is significant but is there a reason why the basal level is so low?

Response #9: We appreciate the reviewer's comment. We believe that the differences in the percentage of BrdU positive cells between normal and transformed BJ cells were due to a relatively slow growth of normal BJ cells as compared to transformed cells, and not due to a significant growth arrest with the control siRNA. To avoid any confusion, we indicated this implication on the page 17 in the revised Fig. 2 legend.

Comment #10: In Supplementary Fig 5a CETSA melt curves the y axis is labelled non-denatured protein fraction but all the curves are 0-1, is this correct? Should the control treatment be the same for all of the samples i.e. the blue curve similar for each plot? There seem to be significant differences for example with the Q9H5X1 sample. The authors also suggest that the plot for arsenite methyltransferase is not good due to a high plateau. This also seems to be the same with the B4DN88 sample (with a higher DMSO control curve). Also there seem to be minimal differences between the control and experimental curves for the last three plots Q13011, P36551 and Q8N684-3 – are these significant?

Response #10: we appreciate the reviewer's comment. We feel it is important to note that these measurements are the amount of protein that remains in the soluble fraction after a heating step (to the temperature on the y axis) and a centrifugation step. This amount is proportional to the amount of folded protein, i.e. non-denatured protein, which is why these curves are melting curves. The amount at 37 °C is put to 1 and it is 0 (zero), when all protein has denatured (and subsequently precipitated). The two curves in each plot for DMSO samples should be as similar as possible but different proteins have different accuracy in the measurements (i.e. depending on the number of peptides that are measured in the MS experiment) and the equation used to select the hits take these variations into account. The equations are however not perfect and we have de-prioritised arsenite methyltransferase when the melting curves of a131 are relatively flat, that is, the shift is more vertical than horizontal, but also because the red curves with a131 appear noisy. We know from our previous experiences, that in the most robustly responding proteins, the melting curves moves

horizontally rather than vertically. The proteins further down in the plot (e.g. B4DN88, Q13011, P36551 and Q8N684-3) are less likely to be significant as the shifts are small or/and curves noisy, but we show these in the Supplementary figures as they are ranked to allow the reader to make a judgment. To further clarify this point, in the revised Supplementary information, we added following sentences to explain about Euclidean distance (ED) score.

Euclidean distance (ED) score of thermal shifts of all the proteins with complete replicates were then calculated as follows:

$$ED \text{ score} = \frac{\sum ED_{\text{inter-treatment}}}{10 \sum ED_{\text{inter-replicate}}}$$

where $\sum ED_{\text{inter-treatment}}$ is the sum of inter-treatment Euclidean distance indicating the apparent shift of compound-treated protein melting curve from DMSO-treated one, while $\sum ED_{\text{inter-replicate}}$ is the sum of inter-replicate Euclidean distance indicating the noise in similarity of protein melting curves from the same treatment in replicate runs. By definition, larger EDS corresponds to the proteins reproducibly showing a significant shift when compared to control, thus they are potential protein hits targeted by the compound or treatment.

Comment #11: In Fig 2b the plots for a131/a166 PI5P4K α and PI5P4K γ seem to be duplicates from the data presented in Supplementary Fig 5. Should the curve for a166 PI5P4K β also be included?

Response #11: We agree with the reviewer's comment. As the reviewer suggested, we added the curve for a166 PI5P4K β in the revised Fig. 2b.

Comment #12. The video file "Mitotic progression of normal BJ cells treated with a131" does not seem to work (downloaded from zip or independently).

Response #12: We apologize for this technical issue. We re-uploaded the video file describing "Mitotic progsuppression of normal BJ cells treated with a131".

Comment #13: In conclusion this is a very well-constructed paper that has an important message, and by addressing the above points would be suitable for publication.

Response #13: We appreciate the reviewer's favourable and constructive comment. The other reviewers also stated every positive comments on our manuscript for publication in Nature Communications. For instance, the reviewer #1 stated in his/her general comment, "In essence the paper is a very interesting study and deserves to be published here if the following criticisms can be addressed". Similarly, the reviewer #2 pointed out, "The validation work is extensive and well executed, and the suggested mechanism is well

elucidated and novel. I am in general in favor of publication after some issues are addressed". We agree with all three referees very favourable comments. In conclusion, we believe that virtually all comments made by all three referees have been experimentally addressed and carefully discussed in the revised manuscript. Therefore, we hope very much that this paper is now acceptable for publication in Nature Communications.

Reviewers' comments:

Reviewer #1 (Remarks to the Author):

the authors have dealt with many aspects of my review. i have a couple of things that are still related to the original review that would need clarifying.

1. the authors state "1). Of note, the difference in GI50 values between normal and cancer 96 cells using MTT assay that measures cell proliferation rate (Fig. 1b & 1e) is likely 97 underestimated, since a131 preferentially induced cell death in transformed and cancer cells, 98 whereas it only arrested normal cells at the G1/S phase of the cell cycle in a transient and 99 reversible manner. Nonetheless, these data suggest that a131 is a potent antiproliferative agent 100 with a clear selectivity toward cancer cells killing.". while i accept this is the case it still is not clear why the authors chose not to just measure apoptosis rather than carry out an MTT test. it is relatively important as most of the manuscript is devoted to BJ fibroblasts that are transformed, and although the system is nice tumor cells are not derived naturally from fibroblasts. therefore the demonstration that this compound has a strong therapeutic window in epithelial derived tumor cells is important. can the authors just redo the analysis and assess apoptosis instead? i will defer to the preference of the editors on this point.

very minor. please change the phrase "inhibited combined cellular PI5P4Ks activity". the assay is solely measuring the in vitro activity of PIP4K not the cellular PIP4K activity

in this respect i would also suggest that the nomenclature for the enzyme be changed to PIP4K (PIP4k2A, PIP4K2B and PIP4K2C) to reflect the gene name. this will inevitably stop further confusion.

in my opinion the manuscript should be accepted after minor revisions.

Reviewer #2 (Remarks to the Author):

I am happy with the revisions, all my concerns have been addressed.

Reviewer #3 (Remarks to the Author):

With reference to the revision of manuscript NCOMMS-17-02097A, "Dual Blockade of the Lipid Kinase PI5P4Ks and Mitotic Pathways Leads to Cancer-selective Lethality" authors Kitagawa et al., provided by Dr Sang Hyun Lee, improvement has been made by consideration of all of the reviewer's original comments.

In the specific response to my comments it is stated that "a series of additional analysis" has been performed to address the remaining issues. I do not see significant evidence of this, however the authors have addressed most of the points either by editing the text or rearranging figures, and have given sufficient explanation to cover these.

Specific comments regarding the rebuttal responses;

Points 2, 3, 6 and 11 deal with text mistakes and unclear figure items and have been acceptably resolved.

Point 12 regarding the video file has been resolved and the additional video clips are supportive of the author's claim and add significant relevance.

Point 9 is adequately dealt with in the revised figure legend.

Point 7 has been addressed by removal of Figure 4j and the text relating to the claim that mechanistically the observations can be due to changes in nuclear lipid levels. Further investigation of this point would have substantially increased the significance of the paper, but I agree that this is probably beyond the scope of the current study.

Point 5 has been adequately explained and referred to in the revised text.

For further consideration;

Point 4 is a fair reflection on the experimental protocol and it would be useful to include this in, for example, the figure legend.

Point 10 is well explained and the additional text regarding selection criteria for thermal shift is adequate. However the y-axis label is not descriptive (it could equally be "denatured protein fraction") and would benefit further explanation in the specific materials and methods section, or better labeling. This axis is inconsistently labeled "stability" in Figure 2.

Regarding Point 1, the rearrangement of the Figures is an improvement, and I thank the authors for their detailed explanation. I agree that the evidence shows that a131 causes growth arrest in normal cells and cell death in transformed cells, the major point of the paper. The legend for Figure 1 is particularly sparse on information and doesn't help with accessibility to a wider readership (reviewer 2 comment 1). Descriptions for Figure 1 c and d are minimal and assume specialist knowledge. The description for the relocated Figure 1 b states that cell viability (ie ability to survive) was determined by MTT assay, hence a reduction in cell viability would imply cell death. These points need to be clarified as originally requested.

In response to point 8 the authors stated that they corrected the poor curve fitting in Figure 2c and 2d. With reference to Figure 2c, the curves for a166 and a131 do seem to be slightly improved. Two additional data points for the control I-OMe (a non-specific PI5P4K α inhibitor) have been included to complete this curve (no error bars – single point?). However the curve for a132 does not remotely fit the data – have the correct criteria been attributed to this attempted fit? The authors suggest that "against purified PI5P4K α , a132 showed some measurable inhibition but only at the high concentration of >10 μ M". This is true based on the wrongly fitted curve, however it seems that at around 2 μ M this compound shows about 25% inhibition based on the actual data. On Figure 2d the a131 curve has been fit to give a maximal 100% inhibition but this is clearly not fit to the two highest data points which somehow show > 100% inhibition. The explanation that correlating in vivo and in vitro PI5P4K activity might be complicated is weak; I don't see how the anti-mitotic activity of a159 alters cell cycle progression, and hence affects the intrinsic activity of PI5P4Ks after cell lysis. Effectively, due to the differences in intrinsic activities of the three isoforms the assays will be measuring the same thing; PI5P4K α activity. The likelihood is that the difficulty in explaining these observations may be due to additional off-target/multiple pharmacological effects and the text should be amended to suggest this possibility. I agree that the authors have other evidence to support the genetic data regarding these targets, but this section of the paper needs to be improved.

Overall the changes have strengthened the case for the paper, and, as is usual, have generated more questions that further ongoing research will need to address. The present form of the manuscript has, in my opinion, addressed the claims that the authors make and, with the provision of the few minor changes suggested, should be positively considered for publication by the editors.

Responses to Reviewers' Comments:

Reviewer #1 (Remarks to the Author):

The authors have dealt with many aspects of my review. I have a couple of things that are still related to the original review that would need clarifying.

Response to the general comment: We appreciate for the reviewer's very positive comment on our revised manuscript. We agree with the reviewer's additional comments. We have carefully clarified the remaining issues and revised the manuscript accordingly.

Comment #1: The authors state "1). Of note, the difference in GI50 values between normal and cancer cells using MTT assay that measures cell proliferation rate (Fig. 1b & 1e) is likely underestimated, since a131 preferentially induced cell death in transformed and cancer cells, whereas it only arrested normal cells at the G1/S phase of the cell cycle in a transient and reversible manner. Nonetheless, these data suggest that a131 is a potent antiproliferative agent with a clear selectivity toward cancer cells killing.". While I accept this is the case it still is not clear why the authors chose not to just measure apoptosis rather than carry out an MTT test. It is relatively important as most of the manuscript is devoted to BJ fibroblasts that are transformed, and although the system is nice tumor cells are not derived naturally from fibroblasts. Therefore the demonstration that this compound has a strong therapeutic window in epithelial derived tumor cells is important. Can the authors just redo the analysis and assess apoptosis instead? I will defer to the preference of the editors on this point.

Response #1: We appreciate the reviewer's comment and we agree with the reviewer's suggestion to directly measure cell death and apoptosis. We would also like to note that to obtain the data presented in the original and revised Figure 1e in our original and revised manuscripts, we analysed a large number of cell lines that include 30 different cancer cell lines and 5 different normal cell lines as listed in the original and revised Table S1. We have discussed this matter with the editor and he/she suggested that we should provide further support for our original statement of "cancer selective cell death" by testing a131 in additional cell lines representative of both normal and cancer cell. Thus, as the reviewer and the editor suggested, we performed the experiments of measuring cell death and apoptosis using several representative normal and cancer cell lines, and the results are presented in the revised Supplementary Fig. 2b. Indeed, a131 dramatically induced cell death only in the cancer cell lines, but not so or marginally so, at best, in normal cell lines, as determined by PI staining to measure sub-G1 population and Annexin V staining (Revised Supplementary Fig. 2b). These results are consistent with all other data presented in original and revised figures (Original and revised Fig. 1c, 1d; Original and revised Supplementary Fig. 1b; Original Supplementary Fig. 2b). Together, these results strongly suggest that a131 is a potent antiproliferative agent with a clear selectivity toward cancer cell killing. These results also support our original statement that the difference of GI₅₀ values in normal vs. cancer cells using the MTT assay does not fully reveal the robust cancer-specific cell lethality induced by a131, since a131 arrested normal cells at the G1/S phase of the cell cycle, which was transient and reversible after a131 removal (Original and revised Supplementary Fig. 1d). We indicated these new results on page 4 (lines 23-28) in the revised manuscript.

Comment 2: Very minor. Please change the phrase "inhibited combined cellular PI5P4Ks activity". The assay is solely measuring the in vitro activity of PIP4K not the cellular PIP4K activity.

Response #2: We appreciate the reviewer's comment and we agree with the reviewer's suggestion to change the phrase "inhibited combined cellular PI5P4Ks activity". We changed the original statement to "inhibited in vitro PI5P4Ks activity" on the page 7 in the revised manuscript.

Comment #3: In this respect I would also suggest that the nomenclature for the enzyme be changed to PIP4K (PIP4K2A, PIP4K2B and PIP4K2C) to reflect the gene name. This will inevitably stop further confusion.

Response #3: As the reviewer suggested, we have changed the nomenclature of the PI5P4K enzymes to PIP4K (PIP4K2A, PIP4K2B and PIP4K2C) to reflect the gene name throughout the revised manuscript, revised figures and revised supplementary information.

Comment #4: In my opinion the manuscript should be accepted after minor revisions.

Response #4: We appreciate the reviewer's favourable and constructive comment. The other reviewers also suggested our revised manuscript to be accepted for publication in Nature Communications.

Reviewer #2 (Remarks to the Author):

I am happy with the revisions, all my concerns have been addressed.

Response to the general comment: We appreciate the reviewer's favourable comment that all concerns raised by the reviewer have been addressed in the revised manuscript. We hope very much that this revised manuscript is now acceptable for publication in Nature Communications.

Reviewer #3 (Remarks to the Author):

With reference to the revision of manuscript NCOMMS-17-02097A, “Dual Blockade of the Lipid Kinase PI5P4Ks and Mitotic Pathways Leads to Cancer-selective Lethality” authors Kitagawa et al., provided by Dr Sang Hyun Lee, improvement has been made by consideration of all of the reviewer’s original comments.

In the specific response to my comments it is stated that “a series of additional analysis” has been performed to address the remaining issues. I do not see significant evidence of this, however the authors have addressed most of the points either by editing the text or rearranging figures, and have given sufficient explanation to cover these.

Specific comments regarding the rebuttal responses;

Points 2, 3, 6 and 11 deal with text mistakes and unclear figure items and have been acceptably resolved.

Point 12 regarding the video file has been resolved and the additional video clips are supportive of the author’s claim and add significant relevance.

Point 9 is adequately dealt with in the revised figure legend.

Point 7 has been addressed by removal of Figure 4j and the text relating to the claim that mechanistically the observations can be due to changes in nuclear lipid levels. Further investigation of this point would have substantially increased the significance of the paper, but I agree that this is probably beyond the scope of the current study.

Point 5 has been adequately explained and referred to in the revised text.

Response to the general comment: We appreciate the reviewer’s favourable comment and we also agree with the reviewer that the concerns indicated above have been properly addressed in the revised manuscript.

For further consideration;

Comment #1: Point 4 is a fair reflection on the experimental protocol and it would be useful to include this in, for example, the figure legend.

Response #1: We appreciate the reviewer’s constructive comment. As the reviewer suggested, we include the statement “Note that detection of apoptosis by TUNEL and scanning of the entire sections of each slides, which contained both tumors and adjacent normal tissues, was performed in a completely unbiased manner. Moreover, for unbiased quantification, we used 5 images from each sections of >6 sections from each group (bottom).” on the page 17 in the revised manuscript.

Comment 2: Point 10 is well explained and the additional text regarding selection criteria for

thermal shift is adequate. However the y-axis label is not descriptive (it could equally be “denatured protein fraction”) and would benefit further explanation in the specific materials and methods section, or better labeling. This axis is inconsistently labeled “stability” in Figure 2.

Response #2: We appreciate the reviewer’s comment. In order to clarify this issue, we have changed the y-axis labels consistently as “Soluble Fraction” in all the CETSA curves in the revised Figure 2b as well as in the revised Supplementary Figure 5. We believe that this would remove any ambiguity in the labeling.

Comment #3: Regarding Point 1, the rearrangement of the Figures is an improvement, and I thank the authors for their detailed explanation. I agree that the evidence shows that a131 causes growth arrest in normal cells and cell death in transformed cells, the major point of the paper. The legend for Figure 1 is particularly sparse on information and doesn’t help with accessibility to a wider readership (reviewer 2 comment 1). Descriptions for Figure 1 c and d are minimal and assume specialist knowledge. The description for the relocated Figure 1 b states that cell viability (ie ability to survive) was determined by MTT assay, hence a reduction in cell viability would imply cell death. These points need to be clarified as originally requested.

Response #3: We appreciate the reviewer’s comment that the rearrangement of the Figures is an improvement and provided with our detailed explanation. We also agree with the reviewer’s comment that descriptions for Figure 1 c and d are minimal and assume specialist knowledge. As the reviewer suggested, we have revised in the Figure 1c legend to “(c) FACS analysis using BrdU and PI double staining of the indicated cells. Percentages of BrdU positive (S) population indicating cell cycle arrest and subG1 (<2N) population indicating cell death are shown”. We have also revised the Figure 1d legend to “(d) Immunoblot analysis of cleaved PARP and caspase-3 (Cas-3) indicating a131-induced apoptosis in transformed BJ cells (lanes 5, 6), but not normal counterparts (lanes 2, 3)”.

As the reviewer also pointed out, we have revised the Figure 1b legend to “(b) Normal and transformed BJ cells were treated with a131, paclitaxel or nocodazole at a range of different concentrations for 72 h in triplicate and cell viability was determined by MTT assay in comparison with indicated antiproliferative agents, hence a reduction in cell viability would imply cell cycle arrest and cell death. Mean values with ± standard deviation (S.D.) are shown (n=3).” These changes are indicated on the page 17 in the revised manuscript.

In addition, as the reviewer #1 and the editor suggested, we performed the experiments of measuring cell death and apoptosis using several representative normal and cancer cell lines, and the results are presented in the revised Supplementary Fig. 2b. Indeed, a131 dramatically induced cell death only in the cancer cell lines, but not or marginally in normal cell lines (Original and revised Fig. 1c, 1d; Original Supplementary Fig. 1b; Revised Supplementary Fig. 2b). Together, these results strongly suggest that a131 is a potent

antiproliferative agent with a clear selectivity toward cancer cells killing. We indicated these new results and implications with MTT assay on the page 4 (lines 23-28) in the revised manuscript.

Comment 4#: In response to point 8 the authors stated that they corrected the poor curve fitting in Figure 2c and 2d. With reference to Figure 2c, the curves for a166 and a131 do seem to be slightly improved. Two additional data points for the control I-OMe (a non-specific PI5P4K α inhibitor) have been included to complete this curve (no error bars – single point?). However the curve for a132 does not remotely fit the data – have the correct criteria been attributed to this attempted fit? The authors suggest that “against purified PI5P4K α , a132 showed some measurable inhibition but only at the high concentration of >10 μ M”. This is true based on the wrongly fitted curve, however it seems that at around 2 μ M this compound shows about 25% inhibition based on the actual data. On Figure 2d the a131 curve has been fit to give a maximal 100% inhibition but this is clearly not fit to the two highest data points which somehow show > 100% inhibition.

The explanation that correlating in vivo and in vitro PI5P4K activity might be complicated is weak; I don't see how the anti-mitotic activity of a159 alters cell cycle progression, and hence affects the intrinsic activity of PI5P4Ks after cell lysis. Effectively, due to the differences in intrinsic activities of the three isoforms the assays will be measuring the same thing; PI5P4K α activity. The likelihood is that the difficulty in explaining these observations may be due to additional off-target/multiple pharmacological effects and the text should be amended to suggest this possibility. I agree that the authors have other evidence to support the genetic data regarding these targets, but this section of the paper needs to be improved.

Response #4: We appreciate the reviewer's comments and we apologize for not including the error bars for the control I-OMe (a non-specific PI5P4K α inhibitor), which are indicated in the revised Figure 2c.

We also agree with the reviewer's comment that in the original Figure 2d the a131 curve has been fit to give a maximal 100% inhibition, but this is clearly not fit to the two highest data points which somehow show > 100% inhibition. To address this issue, we repeated in vitro PI5P4Ks enzyme activity using a131 and replotted the resulting curve, which is presented in the revised Figure 2d.

Regarding the curve for a132 presented in the original Figure 2c, we noticed an inadvertent error with background correction and curve fitting, which showed ~5-7% inhibition without a132. To resolve this issue, we repeated in vitro PI5P4K α enzyme activity using a132 and replotted the resulting curve, which is presented in the revised Figure 2c. In brief, a132 showed about ~10% inhibition of PI5P4K α enzyme activity at both ~2 μ M and ~5 μ M, and ~40% inhibition at the high concentration of >10 μ M. Thus, we agree with the reviewer's comment that a132 has some inhibitory activity against PI5P4K α in vitro. However, as shown in the original and revised Figure 2c, a131 and a166 possess

significantly higher PI5P4K α inhibitory activity than that of a132. Furthermore, our data indicate that the PIP4K genes appear redundant in inducing cell cycle arrest in normal cells, since knockdown of individual PIP4K isoforms did not show a significant growth inhibition in normal BJ cells (Revised Supplementary Fig. 6b & 6c). Together with the results of a132 that failed to kill either normal or transformed BJ cell lines when these cells were treated with a132 at 5 μ M (Original and revised Supplementary Fig. 4c), our data suggest that ~10% inhibition of PI5P4K α enzyme activity in vitro unlikely exerts meaningful biological effects. Nonetheless, we agree with the reviewer's comment that a132 showed a measurable inhibition of PI5P4K α activity despite less than that of a131 and a166. Thus, we clearly indicated this by stating that "Of note, a132 in Group 4 compounds, which failed to kill either normal or transformed BJ cell lines (Supplementary Fig. 4c), exerted ~10% inhibition of PI5P4K α enzyme activity at both ~2 μ M and ~5 μ M, and ~40% inhibition at the high concentration of >10 μ M., although this inhibition was significantly less than that of a131, a166 or I-OMe-AG-538 (Fig. 2c)." on the page 7 in the revised manuscript.

We also appreciate the reviewer's comment that the likelihood of the difficulty in explaining the observations correlating in vivo and in vitro PI5P4K activity may be due to additional off-target/multiple pharmacological effects. Moreover, we agree with the reviewer's comment, "I agree that the authors have other evidence to support the genetic data regarding these targets". Nonetheless, we agree with the reviewer's comment that the text should be amended to suggest the possibility of additional off-target/multiple pharmacological effects. As the reviewer suggested to improve this section of paper, we clearly stated that "It is also worth noting that a159 in Group 3 compounds with anti-mitotic activity exhibited no inhibitory activity against PI5P4K α in vitro, but it did exhibit some measurable inhibitory activity against the endogenous PI5P4Ks (Fig. 2c & 2d), suggesting a possibility that some of the other proteins in the hit list correspond to proteins that make less but direct physical interactions with the compounds leading to additional off-targets or a result of poly-pharmacological effects." on the page 7 in the revised manuscript.

Comment #5: Overall the changes have strengthened the case for the paper, and, as is usual, have generated more questions that further ongoing research will need to address. The present form of the manuscript has, in my opinion, addressed the claims that the authors make and, with the provision of the few minor changes suggested, should be positively considered for publication by the editors.

Response #5: We appreciate the reviewer's favourable and constructive comment. We agree with the reviewer's comment that overall the changes in the revised manuscript have strengthened the case for the paper and the present form of the manuscript has addressed the claims that the authors make. Similarly, the other reviewers are also satisfied with the revised manuscript and suggested to be accepted after minor revisions, which have been addressed experimentally as well as carefully described based on all reviewers' suggestions. Therefore, we hope very much that this paper is now ultimately acceptable for publication in Nature Communications.

REVIEWERS' COMMENTS:

Reviewer #1 (Remarks to the Author):

This paper should now be acceptable.

Reviewers' comments (NCOMMS-17-02097B):

Reviewer #1 (Remarks to the Author):

Comment: This paper should now be acceptable

Response: We appreciate for the reviewer's comment on our revised manuscript. We agree with the reviewer's comment that this paper should now be acceptable for publication in Nature Communications.